# Salinity causes widespread restriction of methane emissions from small inland waters

Cynthia Soued [1,10], Matthew J. Bogard [1,10] ✉, Kerri Finlay [2,3], Lauren E. Bortolotti [4], Peter R. Leavitt [3,5], Pascal Badiou[4], Sara H. Knox[6,8], Sydney Jensen[2], Peka Mueller[1], Sung Ching Lee [6,9], Darian Ng[6], Björn Wissel[3,7], Chun Ngai Chan[1], Bryan Page[4] & Paige Kowal[4]

Inland waters are one of the largest natural sources of methane ($CH_4$), a potent greenhouse gas, but emissions models and estimates were developed for solute-poor ecosystems and may not apply to salt-rich inland waters. Here we combine field surveys and eddy covariance measurements to show that salinity constrains microbial $CH_4$ cycling through complex mechanisms, restricting aquatic emissions from one of the largest global hardwater regions (the Canadian Prairies). Existing models overestimated $CH_4$ emissions from ponds and wetlands by up to several orders of magnitude, with discrepancies linked to salinity. While not significant for rivers and larger lakes, salinity interacted with organic matter availability to shape $CH_4$ patterns in small lentic habitats. We estimate that excluding salinity leads to overestimation of emissions from small Canadian Prairie waterbodies by at least 81% (~1 Tg yr$^{-1}$ $CO_2$ equivalent), a quantity comparable to other major national emissions sources. Our findings are consistent with patterns in other hardwater landscapes, likely leading to an overestimation of global lentic $CH_4$ emissions. Widespread salinization of inland waters may impact $CH_4$ cycling and should be considered in future projections of aquatic emissions.

Methane ($CH_4$) is a potent greenhouse gas responsible for 16% of current atmospheric radiative forcing[1]. Inland waters are the largest natural source of $CH_4$, emitting 398.1 (±79.4) TgCH$_4$ yr$^{-1}$ [2,3]. However, this number is largely based on measurements performed in solute-poor waters[2,4–6], despite salt-rich systems representing roughly half of the global inland water volume[7] and a fifth of inland water surface area (ref. 8 and references therein). There is clear evidence that salinity, particularly as sulfate ($SO_4^{2-}$), inhibits $CH_4$ production through multiple mechanisms, which may lead to lower $CH_4$ emissions from these

systems[9–12]. The paucity of empirical data from salt-rich inland waters raises questions about our current understanding of aquatic $CH_4$ regulation and about the accuracy of global $CH_4$ emissions estimates.

The salinity of aquatic ecosystems shapes microbial communities[13], in particular the abundance and distribution of methanogens and methanotrophs[14,15]. Methanogenesis is the least energy-efficient carbon (C) mineralization process in the redox chain. An abundance of ions favors more energetically efficient reactions, with sulfate ($SO_4^{2-}$) and iron ($Fe^{3+}$) reducers outcompeting methanogens for labile C substrate

[1]Department of Biological Sciences, University of Lethbridge, Lethbridge, AB, Canada. [2]Department of Biology, University of Regina, Regina, SK S4S 0A2, Canada. [3]Institute of Environmental Change and Society, University of Regina, S4S 0A2 Regina, SK, Canada. [4]Institute for Wetland & Waterfowl Research, Ducks Unlimited Canada, PO Box 1160, R0C 2Z0 Stonewall, MB, Canada. [5]Limnology Laboratory, Department of Biology, University of Regina, Regina, SK S4S 0A2, Canada. [6]Department of Geography, The University of British Columbia, Vancouver, BC, Canada. [7]LEHNA, Université Claude Bernard Lyon 1, 69622 Villeurbanne, Cedex, France. [8]Present address: Department of Geography, McGill University, Montreal, QC, Canada. [9]Present address: Department of Biogeochemical Integration, Max Planck Institute for Biogeochemistry, Jena, Germany. [10]These authors contributed equally: Cynthia Soued, Matthew J. Bogard. ✉e-mail: matthew.bogard@uleth.ca

and hydrogen ions[12,16,17]. The reduction of $SO_4^{2-}$, nitrate ($NO_3^-$), $Fe^{3+}$, and other ions can be coupled to anaerobic $CH_4$ oxidation[18–24], further suppressing $CH_4$ concentrations at high salinity. Salinity can also interact with other key $CH_4$ controls, with organic C availability modulating the $SO_4^{2-}$ inhibition of methanogenesis[12,19,25,26], and nutrient availability changing with salt content due to sorption to sediments[19,27]. Salinity thus integrates multiple pathways of $CH_4$ suppression. The inhibition of $CH_4$ by salinity (especially $SO_4^{2-}$) has been demonstrated in coastal and salt-rich wetlands[10,11,28–32], as well as in experiments that demonstrated suppression of net sediment $CH_4$ production under elevated exposure to NaCl[9,33], suggesting a potentially significant effect of multiple ions on $CH_4$ across salt-rich inland waters. We provide an extensive overview of the mechanisms by which salinity may restrict net $CH_4$ production in inland waters (supporting text S1). Ultimately, the impact of these finer-scale controls on regional and continental $CH_4$ emissions is unknown.

Here we survey 193 aquatic ecosystems, including rivers, lakes, open-water wetlands, and agricultural ponds spanning a wide range in morphometry, hydrology, chemistry, and trophic status in the Canadian Prairies (Fig. S1 and Table S1), a region with the highest density of salt-rich inland waters worldwide[34]. We support this with a peripheral survey of wetland ponds (fewer environmental parameters, with greater spatio-temporal sampling of $CH_4$) plus high-resolution eddy covariance data from two wetlands representative of fresh- and hardwater wetland habitats on the Canadian Prairies. We show that salinity restricts $CH_4$ emissions in small lentic waterbodies, especially through ebullition, leading existing freshwater models to overpredict emissions. Regionally scaled, this inaccuracy is highly relevant to the Canadian national emissions budget, as it is comparable to other components of national GHG inventories (e.g., wastewater treatment plant emissions). We show that salinity is a key driver of $CH_4$, interacting with organic matter (OM) content to shape surface $CH_4$ partial pressure (pCH$_4$). Our findings are consistent with data from other global hardwater landscapes, leading us to conclude that salinity likely downregulates emissions worldwide, and current global budgets likely overpredict aquatic $CH_4$ emissions due to overestimates of emissions from hardwater regions.

## Results and discussion
### Salinity–pCH$_4$ link varies across ecosystem types
Across all primary sites where the full suite of limnological parameters was measured, we found clear differences in the predictors of pCH$_4$ among ecosystem types. We used multiple linear regressions to explore the connection between salinity and $CH_4$ while accounting for influences of other known drivers of $CH_4$ cycling, including temperature[35–37], and common proxies for organic substrate (DOC and TP)[4,5,38,39] (Table 1 and Fig. S2). The effect of salinity on pCH$_4$ varied among ecosystem types, from non-significant in rivers and lakes to pronounced control of pCH$_4$ in small lentic systems (surface area <0.1 km², wetlands and agricultural ponds) (Fig. 1, S2, Table 1). In rivers, salinity, DOC, and nutrient concentrations were highly colinear (Pearson $r > 0.6$), making salt-rich rivers also highly concentrated in organic matter (elevated DOC and nutrient content), and overriding a

potential inhibitory effect of salinity on pCH$_4$. Similarly, there was no significant influence of salinity in larger lakes (surface area > 0.1 km²), whereas ionic concentration was a key pCH$_4$ predictor in small, lentic waterbodies (surface area < 0.1 km²) (Fig. 1a,S2).

Among ecosystem types, contrasting environmental characteristics likely led to differences in salinity–pCH$_4$ relationships (Fig. 1a) between the small lentic systems versus larger lakes and rivers. The importance of salinity in small and shallow waterbodies may reflect the closer connection between surface pCH$_4$ and sediment methanogenesis, directly affected by salinity, compared to larger lakes where pCH$_4$ is more susceptible to water column processes like mixing, oxidation, and pelagic production[40,41]. The concentration of OM, nutrients, and ions often covary due to common catchment sources (e.g., agricultural and urban inputs), hydrological transport, and evapoconcentration[34,42–45]. Accordingly, the highest salinity, DOC, TP, and TN concentrations were found in lentic waterbodies with long water residence time (WRT) favoring solute accumulation, while fast-flowing lotic systems were the most solute-poor (Fig. S1 and Table S1). Site-specific factors can decouple OM cycling and ions, with groundwater inputs, local geology, and atmospheric deposition changing ionic water composition[7,34,46]. The influence of salinity on pCH$_4$ observed in small lentic waterbodies (Fig. 1,S2) has the potential for large-scale impacts as these wetlands and ponds are the most abundant aquatic features in the regional Prairie landscape[34,47,48] and a large source of $CH_4$ globally[2,3].

### Interplay between salinity and OM regulates pCH$_4$ in small lentic systems
We explored the connection between $CH_4$ cycling, salinity, and substrate availability (as DOC content) in small lentic systems (Fig. 1b) in more detail. The positive correlation between pCH$_4$ and the DOC/salinity ratio ($R^2_{adj} = 0.15$, Fig. 1b,S3a) suggests that organic-rich systems (with elevated DOC concentrations) can compensate for the inhibitory effect of salinity on methanogenesis. Similarly, pCH$_4$ correlated with ratios of TN and TP to salinity (Fig. S3b,c), indicating that DOC was a proxy for OM content and ecosystem productivity. As an independent validation of this finding, we sampled a peripheral set of 48 wetland ponds across the three Prairie Provinces (Fig. S1), covering the same environmental gradients, but each was sampled repeatedly through the 2021 ice-free season. This independent survey confirmed that the relationship between pCH$_4$ and the ratio of DOC/salinity found here holds at broader spatial and temporal scales (Fig. S4). In line with previous research, the correlation with DOC likely reflects the absence of competition for organic substrate between methanogenesis and other reduction pathways when OM is abundant or when the low concentration of other electron acceptors limits alternative redox processes[12,19,25]. Conversely, low DOC/salinity ratios may favor other reduction reactions[12,16,17] that outcompete methanogens for the limited available OM, enhance anaerobic oxidation of $CH_4$, and restrict net $CH_4$ production through other mechanisms (supporting text S1) thereby lowering pCH$_4$. This inhibitory effect is likely mostly caused by $SO_4^{2-}$ ions in the wetland systems, which correlated strongly with salinity (Figs. S5 and S6), in line with previous studies of brine

**Table 1 | Summary of multiple linear regressions performed on log$_{10}$ transformed (non-standardized) values for the small lentic systems (wetlands and ponds) and for each type of system separately**

| Systems | Equation | p-value | $R^2_{adj}$ |
|---|---|---|---|
| River | Log$_{10}$ pCH$_4$ = 0.18 + 0.34 log$_{10}$ DOC + 0.16 log$_{10}$ Salinity + 0.76 log$_{10}$ TP + 0.05 Temperature | « 0.001 | 0.76 |
| Lake | Log$_{10}$ pCH$_4$ = 10.4 − 1.6 log$_{10}$ DOC + 1.0 log$_{10}$ Salinity − 0.17 log$_{10}$ TP − 0.27 Temperature | 0.28 | 0.14 |
| Pond | Log$_{10}$ pCH$_4$ = 0.47 + 0.57 log$_{10}$ DOC − 0.60 log$_{10}$ Salinity + 0.28 log$_{10}$ TP + 0.01 Temperature | « 0.001 | 0.27 |
| Wetland | Log$_{10}$ pCH$_4$ = 3.5 − 0.50 log$_{10}$ DOC − 0.82 log$_{10}$ Salinity + 0.04 log$_{10}$ TP + 0.02 Temperature | 0.001 | 0.31 |
| Small Lentic | Log$_{10}$ pCH$_4$ = 1.7 − 0.16 log$_{10}$ DOC − 0.56 log$_{10}$ Salinity + 0.33 log$_{10}$ TP + 0.03 Temperature − 0.06 Area | « 0.001 | 0.31 |

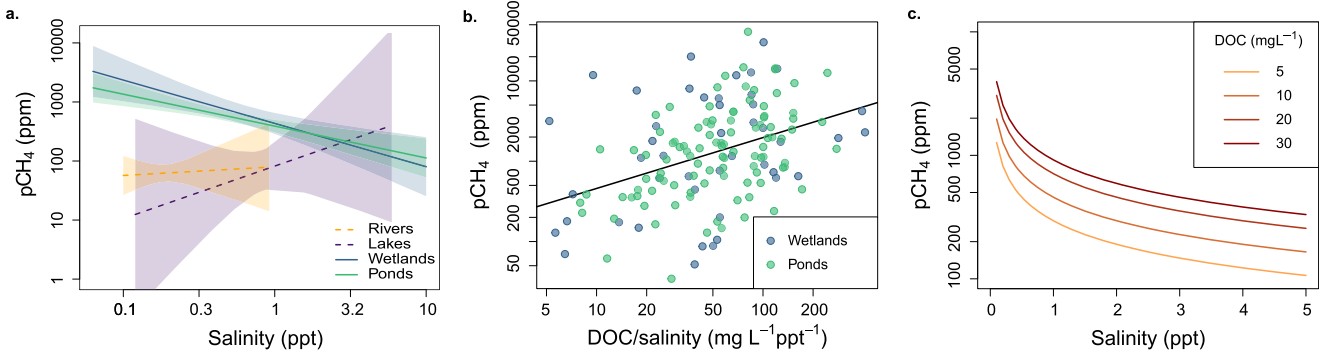

**Fig. 1 | The relationship between pCH₄, salinity, and DOC across ecosystem types.** Marginal effect of salinity on pCH₄ derived from multiple linear regressions (Table 1, Fig. S2), when holding other factors constant at their average value in different system types (**a**). Regressions were non-significant for lakes and rivers (shown as dashed lines). Shaded areas represent the 95% confidence interval around the slope. **b** For small lentic systems (ponds and wetlands), pCH₄ (ppm)

scaled positively with the DOC (mg L⁻¹) to salinity (ppt) ratio ($p$-value « 0.001, $R^2_{adj}$ = 0.15; linear regression equation with standard errors: $\log_{10} (pCH_4)$ = 2.03 (±0.22) + 0.63 (±0.12) $\log_{10}$ (DOC/Salinity)). **c** For small lentic ecosystems, modeled pCH₄ is shown as a function of salinity for varying levels of DOC, based on the empirical equation model from panel **b**.

composition in Canadian Prairie waters[34]. For the agricultural ponds, the relationship between pCH₄ and $SO_4^{2-}$ concentration was considerably weaker than for that with salinity (Fig. S6), indicating additional mechanisms independent of ambient $SO_4^{2-}$ content were relatively more important in restricting net CH₄ production (as detailed in supporting text S1). Despite these differences, salinity as a predictor better captured the diverse mechanisms that may restrict pCH₄ among small lentic ecosystems.

Using the empirical relationship in Fig. 1b, the modeled response of pCH₄ to changes in salt content is non-linear, with the greatest sensitivity at the low end of the salinity gradient (Fig. 1c). Most sampled sites (78%) in the Prairie landscape exhibited moderate salinity <1 ppt (Fig. S1, Table S1), a range within which even modest salinity changes from year to year[45,49] could have a substantial effect on CH₄ dynamics (Fig. 1c). For example, at the same DOC concentration (10 mg L), a 0.5 ppt increase in salinity at low salinity conditions (0.5 ppt) results in a 35% reduction of pCH₄, (710–458 ppm) while a 0.5 ppt increase in salinity from 3.5 to 4 ppt only decreased pCH₄ by 8% (207–190 ppm) (Fig. 1c). The non-linear and enhanced effect on pCH₄ at the lower end of the range of salinity in our simulation is important, because it shows that increased salinity in solute poor systems may have a stronger effect on CH₄ dynamics than in already saline systems. Where anthropogenic pressures result in the salinization of hardwater systems alongside an increased supply of OM and nutrients, the net balance may not be as strong an increase of CH₄ emissions as previously anticipated based on nutrients alone (e.g., ref. 39).

Higher salinity was associated with lower CH₄ emissions from inland waters in the Canadian Prairies, particularly via ebullition from small lentic systems (Fig. 2). Diffusive CH₄ fluxes ranged from 0.11 to 91 mmol m⁻² d⁻¹ in small lentic systems (5 ponds and 5 wetlands) (Fig. 2, Table S1), within the global range reported for small (<0.1 km²) waterbodies[39]. Averaged ebullitive CH₄ emissions rates, measured over the summer months in 10 individual ponds and wetlands (5 each), spanned 6 orders of magnitude, with 70% of measurements lower than the 1st quantile of globally reported ebullition rates for small lentic systems[39]. These systems were each sampled in one central location continuously through June, July, and August, thus providing integrated estimates of CH₄ emissions. Past work has emphasized the need for detailed spatio-temporal sampling to constrain uncertainties in flux budgets for single-system research[50]. We highlight that to constrain regional-scale patterns and gradients, more intensive sampling efforts at each site would have added little to our study, given the multiple orders of magnitude of variability in emissions rates that exist between sites, which is already constrained in our current sampling design

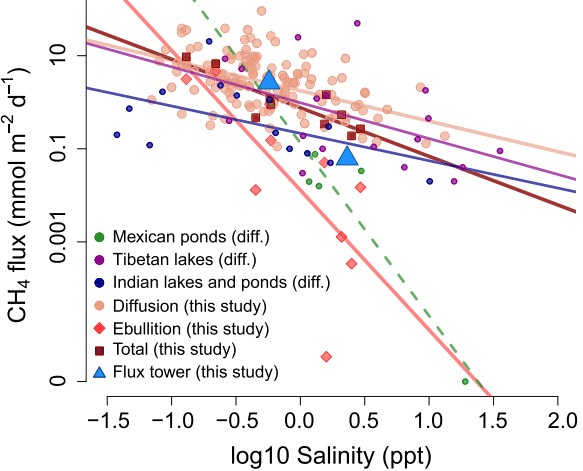

**Fig. 2 | Negative relationship between salinity and aquatic CH₄ emissions in different global salt-rich landscapes.** Significant linear regressions (on $\log_{10}$ scales) exist for diffusion, ebullition, and total flux (ebullitive plus diffusive) from small lentic Canadian Prairie ecosystems (wetlands and ponds; $n$ = 139, 10, and 10; $p$ = «0.001, 0.02, and 0.002; $R^2_{adj}$ = 0.25, 0.47, and 0.69; slope = −0.7, −3.0, and −1.0; intercept = 0.24, −1.9, and −0.12, respectively), for diffusive fluxes from lakes of the Tibetan Plateau[84] ($n$ = 18; $p$ = 0.04; $R^2_{adj}$ = 0.19; slope = −0.8; intercept = −0.01), and for diffusive flux from southern Indian ponds and lakes[85] ($n$ = 14; $p$ = 0.06, $R^2_{adj}$ = 0.2; slope = −0.58; intercept = 0.48). No statistics are presented for diffusive fluxes from the Mexican ponds of the Cuarto Cienegas Basin ($n$ = 5)[86] since one of the values was zero and could not be log-transformed (dashed line). Median emissions values from two Canadian Prairie wetland sites with eddy co-variance towers are also shown.

(see supporting text S2 and Tables S3, S4 for a demonstration of this rationale). Both diffusive and ebullitive flux rates were negatively correlated with salinity (Fig. 2), although the regression slope was much steeper for ebullition than for diffusion (−3.0 vs. −0.7, respectively, on a $\log_{10}$ scale), with a 10-fold increase in salinity (0.1–1 ppt) leading to a 4.5 vs. 1000-fold decline in diffusion vs ebullition, respectively. As detailed in supporting text S1, the stronger influence of salinity on ebullition likely reflects diverse controls that include an impact on sediment CH₄ dynamics (CH₄ production and anaerobic oxidation), whereas water column CH₄ production, consumption, and physical mixing influence surface diffusion and weaken the effect of salinity on this flux pathway.

The extensive monitoring of two wetland sites with eddy covariance towers suggests that the link between salinity and CH$_4$ emissions holds at an ecosystem-level and over annual scales. Emissions patterns from both sites aligned with our regression results (Fig. 2). The two wetlands have diverging salinity levels (0.5 vs. 2.3 ppt) and exhibited two orders of magnitude difference in CH$_4$ flux rates (Fig. 2, S7) when integrating all flux pathways (diffusive, ebullitive, and plant-mediated) and over multiple temporal scales (sub-hourly to annual). Mean and median CH$_4$ emissions at the low salinity site were 20.7 and 8.7 mg C m$^{-2}$ d$^{-1}$ and at the high salinity site were 0.96 and 0.07 mg C m$^{-2}$ d$^{-1}$, respectively. Were salinity not a controlling factor, we would have anticipated consistently high rates of CH$_4$ emissions from both wetland ecosystems, given the hypereutrophic status of both sites and the DOC-rich surface water environments (Table S4). In addition to the role of salinity, differences in CH$_4$ emissions at each wetland may be partly related to other ecosystem features (e.g., open water extent and vegetation features). Future work is needed to fully explain the finer-scale mechanisms and interplay between individual drivers that have led to the large difference in emissions observed here. Overall, our eddy-covariance measurements provide independent support for the Prairies-wide inverse relationship that we observed between salinity and CH$_4$ emissions.

Our meta-analysis showed that the inverse relationship between salinity and CH$_4$ emissions for lentic inland waters exists across the three additional global regions where data were available. Analysis of published data from lakes and ponds from Tibet, Mexico, and India (Fig. 2) show inverse relationships between emissions and salinity. Individually, these relationships are built from limited available data and are of varying strength, but collectively follow a similar trend as the more extensive datasets from the Canadian Prairies (Fig. 2). This further suggests that salinity may restrict aquatic CH$_4$ emissions from hardwater landscapes worldwide (see supporting text S1 for more detail).

## Improving the prediction of emissions from salt-rich inland waters

Existing empirical models[5,36-38] predicted a portion of the variability in CH$_4$ dynamics in a subset of our primary study systems, and for rates of diffusion and ebullition in particular, deviations from predictions were linked to salinity (Fig. 3). The models we tested were developed based on regional and global data largely from freshwater systems[5,36-38], and use typical CH$_4$ drivers including water temperature, lentic surface area, chlorophyll $a$, TN and DOC concentrations. These models captured a third of the observed variability in pCH$_4$ in our dataset ($R^2 = 0.32$; Fig. 3a), and weakly predicted diffusive and ebullitive emissions from small lentic systems ($R^2 = 0.15$ and 0.03 respectively; Fig. 3b,c). Salinity, which is not included in these models, explained 20%, 24%, and 42% of the deviation from predicted pCH$_4$, diffusion, and ebullition, respectively (Fig. 3d-f). Model deviation was large for ebullitive fluxes in wetlands and agricultural ponds, being on average ~3 orders of magnitude lower than predicted (Fig. 3c,f; in line with results from Fig. 2) reflecting a particularly strong impact of

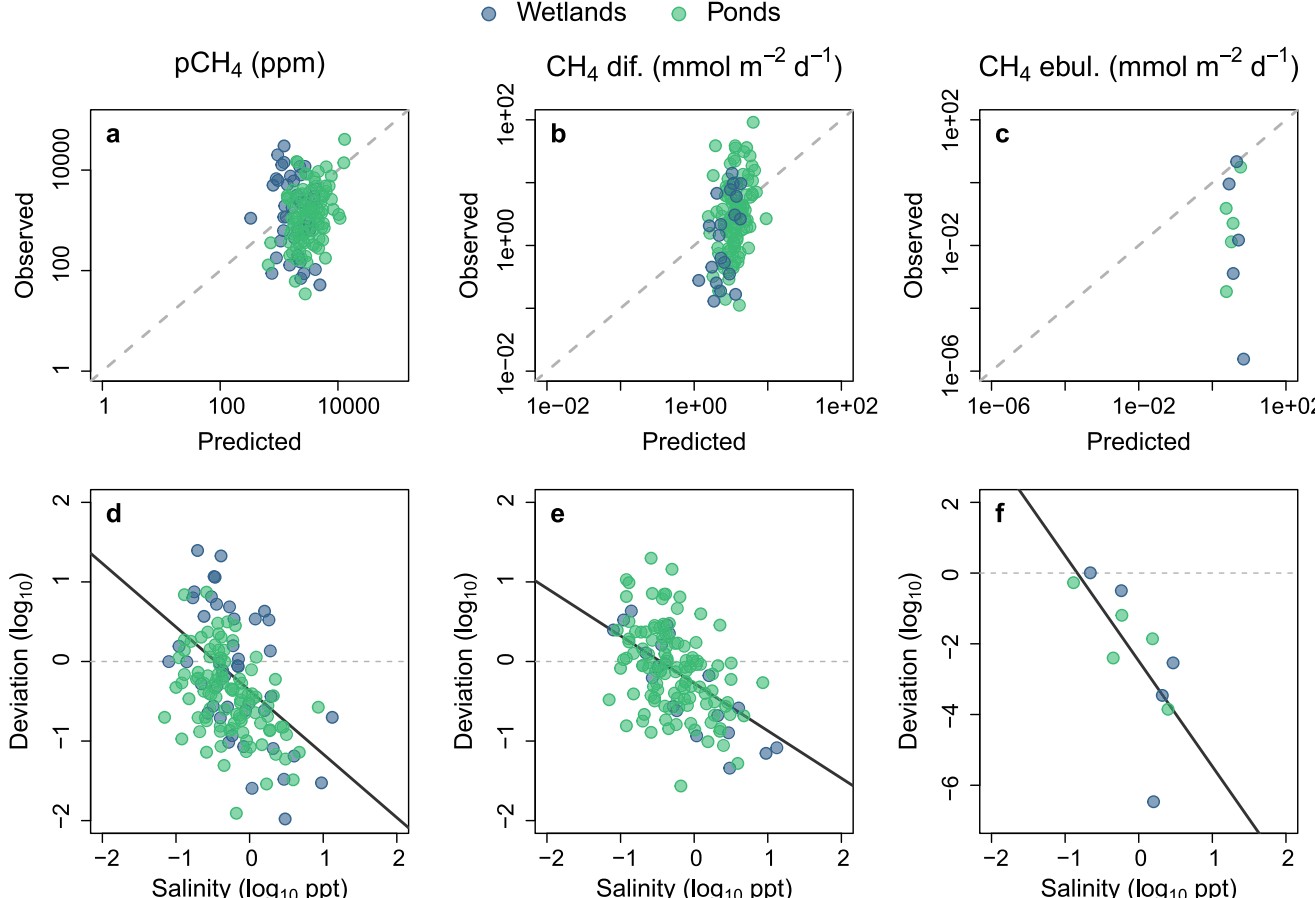

**Fig. 3 | Testing existing empirical models in hardwater Prairie systems.** Observed versus predicted (based on published models[5,36,37]) values of CH$_4$ partial pressure (**a**), diffusion (**b**), and mean summer ebullition rates (**c**) in small lentic sampling sites. The corresponding deviations from predictions (measured - modeled) are shown as a function of salinity (**d**–**f**). The gray dashed line represents a perfect (1:1) correspondence between predicted and observed values. Linear regression lines (solid black) have $p$-values = $6.6 \times 10^{-9}$, $5.4 \times 10^{-10}$, and 0.025, $R^2_{adj}$ = 0.20, 0.24, and 0.42, and slopes = −0.80, −0.60, and −3.0 for panels **d**–**f**, respectively.

salinity on sediment-related processes. Given the generally weak predictive strength of such empirical models, the inclusion of salinity as a model term represents a simple, but major improvement to our ability to predict aquatic $CH_4$ emissions in Canada (Fig. 3) and likely abroad (Fig. 2).

## Salinity impacts large-scale $CH_4$ budgets

By accounting for salinity, the emissions budget for the Canadian Prairies shrinks dramatically as compared to scenarios relying on soft-water model predictions. As a conservative first-order calculation of the potential magnitude of this overestimation, we tested the effect of two salinity levels (0.5 vs. 0.1 ppt). These levels, respectively, correspond to the median salinity value of the small lentic sites that we sampled (Table S1), versus that of typical freshwater systems. We predicted the total $CH_4$ emission rate (diffusion + ebullition) using the relationship from a subset of our primary sampling sites in Fig. 2 ($n = 10$, small lentic sites sampled over three summer months[51]). This yielded a large difference in flux rate (1.57 vs. 8.42 mmol m$^{-2}$ d$^{-1}$, respectively), consistent with the ~3-fold overestimation of total flux rates by existing freshwater models (Fig. 3). The results are within the range of the differences between salt-rich and freshwater sites in our eddy-covariance measurements (Fig. 2, S5). We conservatively applied this simulated difference in emissions to three summer months and used only the areal estimate of small open-water lentic systems in the Canadian Prairies (<0.1 km$^2$, 2869 km$^2$ in total area). This areal estimate is derived from recently completed mapping and represents the best available current estimate of small lentic systems (see the "Methods" section) and is thus a significant advance to scaling Canadian Prairie aquatic emissions. While not included in the scaling estimates, simulations using our peripheral sampling dataset indicated that uncertainties in this cross-province extrapolation stem mostly from spatial coverage, and not enhanced within-site or temporal coverage (Supporting Text S2). This first-order extrapolation exercise resulted in >5-fold lower emissions in the actual salt-rich vs. freshwater scenario (6555 vs. 35,169 MgCH$_4$ over the 3-month period). This difference (0.97 Tg $CO_2$ equivalents) is regionally significant, equal in magnitude to ~11% of the total C footprint ($CO_2$ + $CH_4$ emissions) of the entire beef cattle industry of the province of Saskatchewan[52], or to the entire national emissions budget of the Canadian wastewater treatment sector[53]. Further, the discrepancy in $CH_4$ emissions in our two calculations is roughly equivalent to the entire $N_2O$ budget (in $CO_2$ equivalents) for all US-based Prairie wetlands[54]. Therefore, as a first-order approximation, our calculation demonstrates the importance of obtaining accurate emissions data for Prairie ecosystems in the context of our national GHG emissions inventory. Inaccuracies in these calculations have far-reaching implications for national emissions mitigation, and how aquatic ecosystems are represented in these budgets. The inclusion of salinity represents a simple but major refinement to estimates of aquatic emissions from hardwater ecosystems in Canada and likely other nations (Fig. 2).

Globally, the use of soft-water-derived models to estimate hardwater $CH_4$ fluxes can lead to an overestimation of emissions, with consequences for planetary budgets. We do not presently have the data required to determine the rate of $CH_4$ emissions from salt-rich inland waters at the global scale with confidence. However, our observations from a large area of Canada and three other global regions imply that $CH_4$ emissions are restricted in salt-rich lentic waterbodies, and these fluxes have likely been overestimated by the traditional methodologies used to scale lentic emissions rates to the globe[2]. As a first step to resolving this issue, we demonstrate the potential magnitude of this overestimation at the global scale using the same exercise that we applied to the Canadian Prairies, i.e., we compare the difference in scaled fluxes when using $CH_4$ emissions rates of 1.57 vs. 8.42 mmol m$^{-2}$ d$^{-1}$ (derived from our mid-salinity and

freshwater estimates, respectively). Approximately 12.8% of the global area of small-sized waterbodies are salt-rich (166,120 km$^2$, detailed in the "Methods" section). Using the same conservative 3-month window of time as above, the difference in emissions estimates, when each flux rate is scaled, is 0.38 vs. 2.04 TgCH$_4$ yr$^{-1}$, respectively. This 1.7 TgCH$_4$ yr$^{-1}$ difference represents a potential overestimation equal to 4% of the most recent estimate of global lentic $CH_4$ emissions (41.6 TgCH$_4$ yr$^{-1}$; ref. 55). Using a comprehensive, annual-scale approach and including larger ecosystems would lead to an even larger discrepancy in actual versus currently estimated emissions for hardwater ecosystems. The global lentic ecosystem emissions budget is continually being refined[55], and future explicit consideration of salt-rich inland waters may help to constrain this budget and resolve some of the discrepancies between top-down and bottom-up global $CH_4$ emissions estimates[56].

## Future salinization may impact aquatic emissions

Worldwide, salinization of inland waters is on the rise due to anthropogenic activities (mining, agriculture, urbanization, atmospheric deposition, climate change)[32,42,44,57–62], which may further reduce aquatic $CH_4$ emissions[32]. In the Canadian Prairies, $SO_4^{2-}$ concentrations have increased in 64% of the 14 monitored lakes for which data are available in Saskatchewan over the past 30 years (Fig. S8). This is consistent with previous findings that, despite high-local variability, many Prairie waters are becoming more saline[42,63]. This may in part be linked to rapid increases in the application of S-based fertilizers in agricultural landscapes[64], which may in turn lead to enhanced $SO_4^{2-}$ content and salinity in receiving waters (though such a mechanistic link remains unconfirmed). While future global increases in temperature, eutrophication, and DOC content are expected to stimulate $CH_4$ production[36,65], the effects may be partially counteracted by increases in salinity. To derive more accurate predictions, future emissions scenarios should consider both ambient salinity, and projected changes in ecosystem salinity alongside other variables. Our study demonstrates that ignoring the link between salinity and $CH_4$ emissions can have major implications for the emissions budgets of individual nations where hardwater ecosystems account for an important fraction of regional surface water.

## Methods
### Sampling sites in primary and peripheral surveys
The study was conducted in the Canadian Prairie ecozone of the provinces of Alberta, Saskatchewan, and Manitoba, covering 467,029 km$^2$ (Fig. S1). The region is characterized by a cold continental to semi-arid climate[66], with extreme seasonal temperature variability (monthly means from -19 °C in July to −−18 °C in January[34]), low annual precipitation, and strong winds[67]. Land use is largely agricultural, containing >80% of Canada's farmland area[68]. The region is known for its generally flat topography[34,67], widespread endorheic drainage basins, and high evaporation to precipitation ratio, making hardwater lentic systems (wetlands, ponds, and lakes) largely dominant in the regional aquatic landscape[34,47,48]. The region exhibits the highest abundance and diversity of salt-rich lakes worldwide[34], mostly driven by elevated concentrations of $SO_4^{2-}$ in lakes and wetlands, with more variable ionic composition in agricultural ponds (Fig. S5, but see supporting text S1)[34].

The 193 primary survey sites (Fig. S1) covered a diverse range of system types and sizes, with 23 sampled rivers, 17 lakes, 45 wetlands (shallow open water), and 108 agricultural ponds (Table S1), which are common features of the landscape[69]. Each site was sampled once in the summer (June–August 2011–2021). Peripheral survey sites included a series of 48 Prairie wetlands (16 in Manitoba, 16 in Saskatchewan, and 16 in Alberta) located in the prairie parkland ecozone that were sampled between May and September 2021. For the most part, peripheral survey sites were sampled a minimum of 3–5 times over the course of

the open-water season, except for a few sites that dried up shortly after the initial sampling round. Additionally, 2 of the wetland sites in Manitoba were equipped with eddy covariance flux towers and were sampled more frequently ($n = 10$). The general environmental features of each site are provided in Table S4.

## Environmental parameters

At each primary survey site, water temperature, dissolved oxygen concentration, pH, and specific conductivity were measured near the water surface using a multiparameter probe. The probes used at each site varied in model but were all either Yellow Springs Instruments (YSI) or Hydrolab DS5 model probes, and all were routinely calibrated before sampling. Salinity was either directly measured by the probe or derived from specific conductivity and temperature using the function 'swSCTp' from the package 'oce'[70] in R[71]. Salinity is reported here as ppt (%), consistent with other limnological analyses, where 1 ppt = 1 PSU. Water samples were collected at each sampling site (at <0.5 m depth) in clean bottles and stored at cold temperatures until analyzed for DOC, TP, and TN concentrations. DOC analyses were performed on water samples filtered to 0.45 µm on site or in the lab (shortly after collection), then acidified (to remove inorganic carbon) before being processed on an organic carbon analyzer (Shimadzu TOC-L or 5000A, or Aurora 1030W). TP was analyzed by spectrophotometry on a Lachat QuickChem 8500 instrument using the standard molybdenum blue technique after persulfate digestion. TN was measured after chemical digestion on an Alpkem analyzer (IO analytical Flow Solution 3100), a Lachat QuickChem 8500, or a Dionex DX600 instrument. For 105 sites, TP and TN analysis were performed on filtered samples (0.45 µm) and thus represent the dissolved rather than the total P and N concentrations. However, where both the dissolved and total concentrations were measured, the dissolved fraction represented the vast majority of nutrient content (mean of 77% and 89% for P and N, respectively). Therefore, we assumed the dissolved fraction to be representative of the total nutrient content in these samples. The small error that this introduces in the data was considered in the interpretation of results, and these assumptions have little impact on our observations over such long nutrient gradients. Chlorophyll *a* concentration was measured in most lentic systems by spectrophotometry after filtration (on GF/F or GF/C filters) and alcohol extraction. Concentration of $SO_4^{2-}$ was measured in 118 sites using a SmartChem 200 discrete analyzer or a Dionex DX600 and ICS2500.

## CH$_4$ partial pressure

CH$_4$ concentration in all samples was determined using the headspace technique, however, since the database is a post hoc combination of data collected independently by different research teams, details of the gas sampling method are not uniform. Yet variation introduced by distinct methodologies is expected to be minor relative to the measured range of CH$_4$ content across sites. Overall, water was collected in an airtight container (140 mL plastic syringe or 1.2 L glass bottle or 160 mL Wheaton glass serum bottle). An air headspace was created inside the sealed container, either on-site or in the laboratory (after preserving the water with potassium chloride). The headspace consisted of ambient air at the sampling site or ultrahigh purity dinitrogen in the laboratory, and the air-to-water ratio varied between 0.05 and 0.25. To equilibrate the gas and water phases, the closed container was shaken for at least 2 min. The gas phase was then extracted and analyzed for CH$_4$ concentration by gas chromatography on a GC Scion 456, a Varian 3800, or a Thermo Trace 1310 instrument, calibrated against multiple standards. The in-situ concentration of CH$_4$ in the water was back-calculated based on the gas solubility (dependent on salinity and water temperature) before and after equilibration, local atmospheric pressure, and the headspace air-to-water ratio.

## Calculations of emissions rates

To link CH$_4$ emissions rates to salinity, we used previously published datasets of surface diffusive flux from a subset of the larger primary survey sites, so that we included 139 wetlands and ponds. Published emissions rates for 101 ponds surveyed in 2017 were taken from Webb et al.[72] and diffusive flux rates for the remaining 38 agricultural ponds and wetlands were taken from Jensen et al.[73]. In both cases, the rates of emissions were estimated using the following equation:

$$F = k\left(C_{eq} - C_w\right) \tag{1}$$

where $F$ is the CH$_4$ diffusive flux rate, $k$ is the air–water gas transfer velocity, $C_{eq}$ is the CH$_4$ concentration in the water at equilibrium, and $C_w$ is the CH$_4$ concentration measured in the water. As detailed in both publications, the parameter $k$ was estimated as the average ($\pm 1$ SD) value of 1.64 ($\pm 1.14$) m d$^{-1}$. This value was derived from floating chamber measurements (10 min duration each, $n = 30$) conducted in 2017, on 3 occasions on 10 of the agricultural ponds[72]. These diffusive flux estimates from small lentic systems were then used to test the applicability of existing empirical emissions models (Fig. 3b, e).

CH$_4$ ebullition flux rates initially published and detailed elsewhere[51] were measured at 10 sites (5 natural wetlands and 5 agricultural ponds) that are a subset of the 38 sites for which we used published diffusive flux data[73]. At each site, bubbles were collected by deploying two inverse funnel traps (0.061 m$^2$ and 27.3 cm diameter), one in the shallow and one in the deep region of the waterbody, sampled from June to August to determine the volume of bubbles emitted. Here we report the seasonally averaged value of ebullition from deep sites because most measurements in shallow sites were lost due to instability of traps in shallow water. The mean value with associated error bars ($\pm 1$ SD) for each site is shown in Fig. S9. The CH$_4$ content of bubbles upon ascension was determined by perturbing the sediments and collecting freshly emitted bubbles with funnel traps fitted with an airtight plastic syringe. Gas collected in the syringe was injected in 12 mL Labco vials and analyzed for CH$_4$ concentration as above. Summer mean CH$_4$ ebullition rate was calculated for each site as the product of the total volume of bubble emitted and the concentration of CH$_4$ in freshly emitted bubbles, converted to an areal daily flux rate (per m$^2$ of surface water)[51].

## Eddy covariance measurements

The eddy covariance method was used to take continuous, ecosystem-scale measurements of CH$_4$ flux over two prairie wetland sites of diverging salinity levels over a full year (May 20, 2021 to May 19, 2022). A flux tower was constructed at each of the sites with sensors mounted at ~4.1 m at both sites. These sensors include a 3-dimensional ultra-sonic anemometer (RM Young 81000, RM Young Inc.), an open path CO$_2$/H$_2$O infrared gas analyzer (IRGA) (LI-7500A, LI-COR Inc.), and an open path CH$_4$ IRGA (LI-7700, LI-COR Inc.). An LI-7550 analyzer unit housing LICOR's Smartflux 2 software was installed to collect and process high-frequency (20 Hz) flux data on-site. High-frequency data was subsequently processed into 30-min averages using the Eddypro v7.0.6 software and then filtered to remove bad data and measurements taken under poor atmospheric conditions.

Using the R package REddyProc v 2.2.0[74], moving point tests were used to determine their minimum friction velocity thresholds for each site[75]. Half-hourly measurements with friction velocities below this threshold were removed for having insufficient atmospheric turbulence to produce good-quality data. Further filtering by stationarity and turbulence conditions was performed through quality checks using the flagging system proposed by Mauder and Foken[76]. Data flagged as "2" were identified as bad-quality data and removed from the dataset. Spike detection and removal with variance thresholds were also conducted to remove bad data. Finally, CH$_4$ fluxes received when LI-7700 sensor signal strength dropped below 20%, and when

winds passed through the flux tower between 330° and 30° wind directions were removed.

Data gaps in meteorological variables were filled by merging data from nearby Environment and Climate Change Canada climate stations, and gaps in $CH_4$ data were filled using the random forest method[77]. The final datasets used for this study represent the filtered and gap-filled data from each prairie wetland site.

## Statistical analyses and mapping

All statistical analyses and map construction were performed in R[71]. Prior to analysis, data were $\log_{10}$ transformed if necessary to meet normality requirements. The links between $pCH_4$ or flux and other variables were assessed via multiple linear regressions using the function 'lm' and verifying the homoscedasticity of the residuals with a Shapiro-Wilk test (function 'shapiro.test'). The same regression analysis was performed on standardized values (to a standard deviation of one) to compare effect sizes (coefficients) of explanatory variables (Fig. S2). A marginal effect analysis was performed for each system type using the R package 'sjPlot'[78] to visualize the effect of salinity on $pCH_4$ when other variables are held constant (at dataset average).

Measured $pCH_4$, diffusive, and ebullitive fluxes were compared to predicted values based on existing literature models. We selected multiple models that use established $CH_4$ predictors measured in our dataset, and that are based on empirical surveys in Canada or at the global scale. Following Rasilo and coauthors, $pCH_4$ was modeled based on water surface area, temperature, and TN concentration for lentic systems (lakes, wetlands, and ponds)[37], using a model based on an empirical linear relationship developed from large-scale surveys in eastern Canada[37] with the partial pressure (in ppm) estimated as $pCH_4 = -0.26 - 0.31(\log_{10}\text{Area}) + 0.03(\text{Temperature}) + 0.66(\log_{10}\text{TN})$. Predicted $CH_4$ diffusive and ebullitive flux rates were calculated as the mean of two individually modeled values. First, we predicted diffusive emissions using a global model that includes water surface area and Chl $a$ concentration[5], with rates in mg C m$^{-2}$ d$^{-1}$ estimated as $\log_{10}CH_4$ emissions $= -0.167(\log_{10}\text{Area}) + 0.53(\log_{10}\text{chla}) + 0.098(\log_{10}\text{Area}) \times (-\log_{10}\text{chla}) + 0.705$. Second, we used a regional East-Canadian model including surface water area and temperature[37], with rates also in mg C m$^{-2}$ d$^{-1}$ estimated as $\log_{10}CH_4$ emissions $= -0.37 - 0.29(\log_{10}\text{Area}) + 0.06(\text{Temperature})$. Likewise, predicted rates of $CH_4$ ebullition were considered as the mean of a global model based on a global relationship with Chl a[5], where $\log_{10}CH_4$ emissions $= 0.758(\log_{10}\text{chla}) + 0.752$. We also estimated ebullitive flux rates from a regional (Eastern Canadian) model that includes sediment temperature as a predictor[36], where $\log_{10}CH_4$ emissions $= -1.19 + 0.11(\text{Sediment Temperature})$. We approximated sediment temperature as our measured surface temperature minus 1 °C. While comparing results to different existing individual models from the literature would be a useful exercise, this was not the objective of our study and instead relied on the use of multiple models to account more completely for the different potential controls on $CH_4$ cycling and reduce the bias associated with any one model.

## Spatial upscaling

To estimate the effect of salinity on $CH_4$ emissions at a regional scale, we simulated the $CH_4$ total flux rate using the regression model developed in Fig. 2 as a function of two salinity levels: 0.5 ppt (median of sampled small lentic sites) vs. 0.1 ppt (typical value in freshwaters). The regression model is based on 10 sites (5 wetlands and 5 ponds) with robust data on both diffusive and ebullitive summer fluxes, providing total $CH_4$ emissions from open-water environments (excluding emergent vegetation). The difference in $CH_4$ flux rate between the two scenarios was applied to 3 months of summer (91 days) and to the regional surface area covered by small open-water lentic waterbodies. We calculated the lentic surface area of the Canadian Prairie ecozone using a combination of geospatial layers: the Canadian Wetland

Inventory (CWI; systems $\geq 0.0002$ km$^2$)[79], and the Alberta Biodiversity Monitoring Institute/Ducks Unlimited Canada Southern Saskatchewan Moderate Resolution Wetland Inventory (ABMI/DUC 2022; systems $\geq 0.0004$ km$^2$)[80] for areas outside of the CWI layer coverage (45.9%). We selected only waterbodies of 'shallow/open-water' type, excluding marshes (which are distinguished from shallow/open water by having >25% emergent vegetation cover) and filtered out riverine wetlands (to exclude floodplains) to get a conservative areal estimate of lentic waterbodies of 8843 km$^2$, representing 1.9% of regional land cover. We restricted the upscaling exercise to lentic systems <0.1 km$^2$ (totaling 2869 km$^2$) to remain in the size range of our empirical model. The product of this regional aquatic area by the difference in $CH_4$ flux rate in hardwater vs. freshwater scenarios represented the potential overestimation of $CH_4$ emissions from small waterbodies of the Canadian Prairies if these systems were considered as salt-poor (0.1 ppt).

As a first-order approximation of the potential salt effect on $CH_4$ emissions at the global scale, we estimated the global area of salt-rich small lentic systems and calculated their emissions based on our two reported emissions rates from the Canadian Prairies (intermediate salinity versus freshwater rates). The area of 166,120 km$^2$ was derived from the tentative lower-bound estimate of the global area of salt-rich inland waters (538,892 km$^2$)[81] multiplied by 0.31, the fraction of small (<0.1 km$^2$) systems based on the global size distribution of lakes[82]. Using this approach, the fraction of salt-rich to the total area of small lentic systems is 12.8%[81,82].

## Temporal trends

To explore temporal trends in $SO_4^{2-}$ concentration in the study region, we used a publicly available (Saskatchewan Water Security Agency), long-term monitoring dataset collected in southern Saskatchewan lentic systems between 1990 and 2020. A subset of sites was selected for long-term trend analysis based on the following criteria: (1) a minimum of 10 observations within the 1990–2020 time period, (2) observations span at least 5 years, (3) at least one recent observation after 2010. Where all three criteria were met, a Sen slope analysis was performed using R package 'zyp'[83] to determine the trends in $SO_4^{2-}$ over the past 3 decades. For each Sen slope, a 95% confidence interval (CI) was calculated and used to determine the significance of the slope, with trends considered significantly increasing or decreasing if the CI range was entirely positive or negative, respectively (Fig. S8).

## Data availability

All data used in this study are openly available from multiple sources. Aquatic biogeochemical data from published articles are openly available as linked to published manuscripts or from the Government of Saskatchewan Water Security Agency by request. New data or published sources from primary and peripheral survey sites are available in the Federated Research Data Repository (https://doi.org/10.20383/103.0848). Eddy covariance and associated data are available at the Ameriflux website by accessing sites CA-EM1 (Newdale Manitoba; https://ameriflux.lbl.gov/sites/siteinfo/CA-EM1) and CA-EM2 (Shoal Lake Manitoba; https://ameriflux.lbl.gov/sites/siteinfo/CA-EM2). Hydrometric data for rivers are openly available at https://wateroffice.ec.gc.ca/search/historical_e.html with corresponding station codes for primary sampling sites listed in supporting data.

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

## Acknowledgements

This study was funded by a MITACS-Accelerate grant with Ducks Unlimited Canada, awarded to C.S. in collaboration with L.E.B. and M.J.B., and a postdoctoral fellowship from the National Scientific and Engineering Research Council (NSERC) to C.S. M.J.B. was funded by awards from the University of Lethbridge, NSERC (RGPIN-2020-05302), Canada Foundation for Innovation (CFI) and the Canada Research Chairs (CRC) program (CRC-2018-00041). P.R.L. was funded by the CRC and CFI programs, NSERC, the Province of Saskatchewan, and the University of Regina. K.F. was funded by the Government of Saskatchewan (grant no. 200160015) and NSERC. L.E.B. and P.B. were supported by Ducks Unlimited Canada's Institute for Wetland and Waterfowl Research. P.B. and S.H.K. were funded by the Beef Cattle Research Council, Ag Action Manitoba, an NSERC Alliance grant (ALLRP 555468-20), and S.H.K. was funded by the CRC program and NSERC (RGPIN-2019-04199). B.W. was funded by NSERC and the Saskatchewan Ministry of Environment Fish and Wildlife Development fund. We thank David Butman and Lisa Windham-Myers for important discussions related to data analysis and interpretation and Llwellyn Armstrong for the analysis of $pCH_4$ variance.

## Author contributions

C.S. and M.J.B. conceived the study, and L.E.B. and K.F. contributed to the ideas and conceptualization of the research. Data collection in the field and laboratory was conducted by M.J.B., L.E.B., S.J., K.F., P.B., S.H.K., S.C.L., D.N., B.P., P.K., P.R.L., and P.M. M.J.B., K.F., L.E.B., S.H.K., P.B., P.R.L., and B.W. provided materials. C.S. and C.N.C. conducted the meta-analysis. P.B., S.H.K., S.C.L., and D.N. collected and processed eddy-covariance data. C.S. and M.J.B. curated all data and analyzed aquatic ecosystem data. C.S. wrote the original draft of the manuscript with support from M.J.B. All authors provided input and participated in the manuscript editing and revision process.

## Competing interests

The authors declare no competing interests.
