## [Peer Review File · Nature Communications]

Reviewers' comments:

Reviewer #1 (Remarks to the Author):

Review of research article “Salinity causes widespread restriction of methane emissions from inland waters”, submitted by C. Soued et al. for publication consideration in Nature Communications.

The key hypothesis of their research is that the exclusion of salinity (namely sulfate, SO_4^{2-}) from methane emission models from inland waters may cause an overestimation of emission estimates. The authors estimate that excluding salinity effects may lead to an overestimate of emissions from small waterbodies by ~80%. This could become more relevant in future climate scenarios if salinity increases due to changes in precipitation and anthropogenic activities. The authors used data collected from inland waters in a Canadian Prairie region, including rivers, lakes, rivers and ponds. Additionally, they present methane emission data from two wetlands of varied salinity using the atmospheric eddy correlation technique. In this case, the saltier system clearly had lower methane emissions than the less salty system. The authors also show that CH_4 ebullitive emissions were reduced in more saline waterbodies. In fact, it seems based on their data and interpretation that sulfate inhibits sediment methane formation, which suppresses fluxes at the sediment water interface – either diffusive or ebullitive. However, the sulfate does not appear to suppress the methane production in the water column of (larger) lakes, which could point to other mechanisms of CH_4 formation, i.e. processes related to “oxic methane production”.

The data set is extensive; however, I had a difficult time following the data, how they were obtained, the environmental setting and the implementation, resulting in multiple readings of the manuscript. I have a few issues (and questions) with the manuscript. The fact that sulfate suppresses methane production (and thus fluxes) is well known (especially in the oceanographic community), but perhaps less known is the number of brackish (i.e. sulfate-rich) lakes – which the authors use in their estimates. However, the authors' key message that emission models should include salinity (that is sulfate, I disagree with using salinity as a parametrization here) could be a solid message, however, it is still unclear how this could be applied, and in which cases.

Throughout the manuscript, I question the use of Salinity as this does not necessarily imply sulfate and could result in considerable error in the authors' correlations. In fact, they measured SO_4^{2-} in a lot of the study sites, so I do not understand why this was not used as the key parameter. While the Fig S3 is

somewhat convincing relating Log of sulfate to Log of salinity, since it is on a log-log scale I suspect there is substantial error here. My point, is salinity really an appropriate measure here? I understand it is the easiest to measure, but perhaps the actual electron acceptor that outcompetes the methanogens would be a much more solid analysis.

The data collection on the various lakes (inland waters) is vague and needs to be expanded. Specifically, the data for the individual lakes/ponds, their properties (depth, area, etc), ancillary data, dates of data collection. I believe these data must anyway be supplied should the paper be published. One concern is that the lakes could be overturning while sampling occurred, which significantly affects surface methane, salinity and DOC – especially if the lake is experiencing an anoxic hypolimnion. In fact, the authors state that each lake was sampled 3 – 5 times, however no statistics or at least mean and STD are given.

A final point, I think that rivers should be excluded in this analysis, as these are completely different systems than lakes and ponds (high flow rate, turnover, rapid surface gas exchange), etc.

My suggestion would be, including the above comments, include only data where sulfate was measured, and only include lakes and ponds. The fact that these two systems respond oppositely in terms of CH₄ to increasing SO₄²⁻ (though the correlation for CH₄ and SO₄²⁻ for lakes is very weak) deserves much more description and investigation. I question the use of CH₄ concentrations in these types of studies, as there are some many parameters that can affect concentration in the surface waters (wind exposure, oxidation rates, light penetration, etc). I think using rates (i.e. fluxes) is much more robust and revealing. This is a nice data set, and potentially nice story. Using only the waterbodies where CH₄ flux, concentrations and SO₄²⁻ are all measured would make for a interesting, and very applicable study – especially when validating the data versus the methane modeling approaches.

I have specific comments below:

The use of “core” study site is ambiguous, as I often think of sediment core. Suggest changing to “primary”.

Line 40 – it would be interesting (if possible) to know how salt-rich lakes are currently represented proportionately in the database used to estimate current global estimates of inland water emissions.

Line 42 – it is very well known that SO₄²⁻ inhibits CH₄ production. Check the oceanographic and coastal research literature.

Line 50 – you list SO₄²⁻ and Fe³⁺ as energetically favorable electron acceptors. How important are other EAs in these systems? The relevant ones should be listed here.

Line 60 – the authors survey 193 freshwater systems. As a reader, I have an issue interpreting these results without more information on the individual waterbodies.

Line 78 – define “full suite of limnological parameters”.

Line 81 – I am not clear why the authors present river systems in this paper, as these systems are substantially different than the ponds and lakes. I would suggest omitting the rivers from this study, and diving deeper into the fact that salinity affects CH₄ in small systems (i.e. ponds), and not lakes. It would be interesting to know where the bathymetric cutoff would be. They touch on this point in line 88, but think that this could be further explored.

Line 94 – It would be helpful to the reader if the authors could present upfront the global representation of wetlands and ponds in general, and the fraction that are brackish – or better yet, the ones with elevated SO₄²⁻. A table here quantifying the global setting would be appropriate.

Line 100 – why not show the relationship between CH₄ and TN/TP ratios? This can easily be added to the supplemental.

Line 204 – the authors state “Existing empirical models” are used to compare the model results with their measured data. As this is such a key component of the study, I suggest that the model details be presented so that the reader doesn’t have to search for them in the literature.

Line 208 the model is driven by chlorophyll a, I would be curious how the data presented here correlate with CHL_a – could this be a main driver?

General comment

Confirm that tables and figures (including supplement) are listed in order.

Figure 1a – This figure is very hard to read. As this is a key figure for the message of the paper, I make the following suggestions. Remove the River data (from the study in general), and present the lake, ponds and wetland data in individual panels with the data behind (mean and std for each site). The correlation for the wetlands and ponds is excellent – it would be great to see with the actual data. It is interesting (while very weakly correlated) why the methane increases with salinity in the lakes. In my opinion, this deserves a more detailed focus within this paper, as it is contrary to the authors’ key hypothesis. That is, if the salinity decreases the methane in ponds and wetlands, is this compensated by an increase in methane production with salinity in lakes?

Figure 1b – Again, I strongly suggest removing rivers from this analysis. Overall, this correlation is very weak and I find this unconvincing. Esp. including all the different water types in a single correlation. I

suggest developing the individual correlation for each waterbody type. This correlation however is the basis for the model shown in Fig. 1c. I question if this can be applied in the manner it has been here. For the first thing, since Fig. 1a shows an inverse relationship between salinity and methane, I am unclear how these data can be used to infer the model. The separate contributions of DOC and salinity in the model in Fig. 1c could be better explained.

Figure S2. This figure shows the effect size and the confidence of the regression analysis on the various parameters driving the concentrations of dissolved pCH₄. It seems that the salinity only negatively impacts methane in the small lentic systems. While the effect size is shown to be large for lake (salinity) methane, it seems to be neutral which I don't fully understand. The strongest effect is the area – which I feel should be discussed in more detail. Statistics aren't my strong point, but I have trouble understanding what this figure actually means. Also, in this figure, the salinity only really seems to have a strong effect on small lentic systems. Also, DOC seems to have the largest (positive) effect on methane, but this is not seen in the individual class systems (i.e. lakes vs rivers vs small lentic). How is this explained?

Methods – Sampling time: How does that effect outcome due to changing climate (June – August).

Sites were sampled 3-5 times, it would be useful to know the range of values and variability in salinity (i.e. sulfate), pCH₄ and DOC during these times.

Methods – Line 147, which type of multiparameter probe? Only surface, or were profiles used?

Methods – Line 393, how was the wind measured at each site? What is the justification for using K600 parameterization from Cole and Caraco, 1998? That study used tracers to estimate k₆₀₀, and has since been shown to underestimate k₆₀₀. My suggestion, since the authors performed direct chamber measurements on 10 ponds, calculate the k₆₀₀ directly from their data and compare with other parameterizations that may be better suited, e.g. MacIntyre et al. 2010 Geo Res Lett.

Methods – Line 398, The authors should provide more details on how bubble ebullition was measured. How were the instruments (funnels) deployed to ensure that they did not disturb the underlying sediments? It would be interesting to see the maps of the ponds, depths of deployment and data.

Reviewer #2 (Remarks to the Author):

The article by Soued, Bogard and others, explores the controls of salinity on methane emissions in inland waters. There are multiple studies showing the importance of salinity on methane emissions, but this study leverages a uniquely large dataset across many systems to provide a landscape level assessment of the role of salinity in freshwater methane emissions. The article is well written and except some issues that I detail below, the analysis is compelling and supports the story well. Below I detail the bigger issue I have with the study and then some minor issues that could be addressed in a future revision.

MAIN ISSUES

The biggest issue I have with this study is how all different aquatic systems (rivers, lakes, ponds and wetlands) are combined in the analysis. The breadth of systems is a strength of the manuscript, but the way it is used may not be fully accurate. This is because in some systems (rivers and large lakes) salinity is not that important relative to other factors according to their analysis, and therefore the conclusions and subsequent analysis should reflect that. The title “Salinity causes widespread restriction of methane emissions from inland waters” promises that salinity is THE driver across all systems and is widespread, while the analysis doesn’t fully support it. I will pull a couple of examples to better illustrate the problem, and then suggest a way to address it.

-In the opening sentence of the results and discussion (L79) states: “Across all core sites where the full suite of limnological parameters were measured, salinity had a strong negative effect on pCH₄ in a multiple linear regression that also included temperature^{33–35}, and common proxies for organic substrate (DOC and TP_{4,5,36,37}) (Table S2 and Fig. S2)”. Here I am not sure if they refer to the multiple linear regression including all data, or the system specific linear regression. If the former, I am not sure is statistically sound to include all these different aquatic systems in the same analysis, without including the system as a factor in the model. If the latter, it is not correct given that the MLR for lakes and rivers show in fact a positive effect of salinity on methane emissions (table S2).

-In the important analysis of Figure 1. Is the regression in panel b on all data, including big lakes and rivers? If so, is it meaningful when they behave so different than ponds in wetlands? The analysis in panel c depends on the relationship in panel b, and given the conflicting patterns between lakes/rivers vs ponds/wetlands (shown clearly as the marginal effects in panel a), it may be a bit misleading to use all data to build a model of DOC/salinity vs CH₄, which is then used to do some nice and clean lines of salinity and CH₄ across DOC (panel c). I don’t dispute these results, but that it may not hold as cleanly for rivers/lakes as for the other systems based on the data provided.

The authors are clear about the fact that salinity is not a first-order control across all systems (lines 83-90), show the data in the multiple linear regressions (Table and figure S2) and the marginal effects in figure 1a. But then in some other cases the authors use all data lumped together to do some broader analysis and claims, that to the reader it may appear that they are applicative across all inland waters

while they are not for larger lakes and rivers (again based on this dataset). I would suggest two ways to address it:

-Solution 1: The authors have done a much more extensive effort on the aquatic systems that have a strong effect of salinity, with a regional survey of ponds and some eddy covariance data for wetlands, great work! Given this, maybe the work should have an initial assessment across all systems as follows: “we sampled all those systems, seems like the effect of salinity is stronger in wetlands and ponds because X, in lakes and rivers its non-significant and even positive because of Y. We then proceed with a detailed analysis and landscape relevance of salinity controlling CH₄ emissions in ponds and wetlands”. This would still provide a compelling work and with a bit more rigour, maybe you should leverage in the introduction then that across all inland waters, wetlands are the largest source of CH₄ and that small ponds the most uncertain one.

-Solution 2: Use literature data to expand the analysis for lakes and rivers. There is a large new database of river methane data that includes conductivity data (<https://essd.copernicus.org/preprints/essd-2022-346/>), and a new global estimate of lakes (<https://agupubs.onlinelibrary.wiley.com/doi/full/10.1029/2022JG006793>) and reservoirs (<https://doi.org/10.1029/2021JG006305>) with supplementary data. This would be more work and I am not sure the lentic products have salinity data, but it could be crossed with other chemical databases. This would be more work, but if you prefer to maintain the focus across all inland waters more data is needed to support the role of salinity in other aquatic systems besides ponds and wetlands.

MINOR ISSUES

-L 41: Instead of freshwater volume, what is relevant in this article is the surface area from an emission perspective.

-L60: Here the number of sampled sites is a bit confusing, I would rephrase to: “Here we survey 193 aquatic systems, including rivers, lakes, open-water wetlands, and agricultural ponds and spanning...”.

-L94: Maybe worth noting that small lentic water bodies are the largest source of uncertainty in aquatic methane emissions, both in rates and the total global surface area.

-L113: Here is unclear what aquatic systems you are speaking about. Is it all of them? Ponds?

-L124-L133: This paragraph is quite short and technical. Maybe it can be clarified a bit. Two suggestions here:

-L128: Instead of giving specific changes in ppm of CH₄, all of that could be simpler by saying “For example, at the same DOC concentration (10 mgL), a 0.5 ppt increase in salinity at low salinity conditions

(0.5 ppt) results in a 50% reduction of CH₄, while a similar increase in modest salinities (4 ppt) only decreased CH₄ by 11% (Figure 1c)”.

Then at the end is missing a take home message. For example a quick draft of one: “Thus, increased salinity in solute poor systems may have a stronger effect on methane dynamics than in already saline systems, given anthropogenic pressures that result in salinization of freshwater systems (ref) but also in an increased supply of OM and nutrients, the net balance may not be an increase of methane emissions as previously anticipated (ref).”

-L150: Ecosystems means ponds here right?

-L176: You could even say that the low salinity wetland has three times more DOC than the high one to further support this result. Take it or leave it.

-L199: For the Mexican pond data, if 0, how is it shown in the plot and the regression line calculated? Also, there is a study of Spanish saline lakes that could be worth to include in this figure <https://www.mdpi.com/2073-4441/9/9/659>

-L204: Here and in figure 3, it is all based on published empirical models but these models are not presented in the results. For transparency, I would include the models in the methods, and how they perform in the supplement materials. These results further strengthen my main issue above, given that the model for rivers does OK (with an offset, but these are different systems from the model) and also for lakes. This analysis would be stronger and more compelling if restricted to ponds and wetlands.

-L177: The regional upscaling is sound and compelling. In the global attempt though, the authors are cautious enough so it is ok, however I find more interesting to highlight that both the area and total emissions are heavily uncertain for ponds, and this study provides an important mechanism that could refine global estimates. This is because global estimates from most inland waters are still quite bad from a mechanistic perspective, so maybe no need yet to quantify biases if salinity is not included.

-L342: Does eddy covariance need to be in caps?

-L 358: Would be helpful to know what was the norm and the exception. Any number to say what % of sites that were filtered or not?

-L367: links or reference to the data sources?

-L369: While still being in Canada, that relationship is from another biome that changes those hydraulic scaling relationships. I would rather use a larger assessment such as Raymond et al 2012 L&O Fluids and Environments (<https://doi.org/10.1215/21573689-1597669>)

-L392: I am not a wetland person, but is it common to use a lake based model of gas exchange for wetlands?

-Table S2: Would be interesting to see here whether each given parameter is significant or not in the regression. Instead of using gray formatting for the whole equation, it could be used for that, or using an asterisk .

-Now seeing fig S1 I see that most rivers sampled are in Alberta, could there be some regional patterns here of salinity in inland waters, rather than a system difference? Just wondering, no need to do a full analysis on this.

-Figure S2: Typo in "Small lent." Top right corner

Responses to Reviewer Comments.

Reviewer Comments:

We thank both reviewers for their careful reading of our paper and thoughtful comments that have guided us in making the paper better. We hope that the explanations below that detail our changes to the manuscript, or justifications/clarifications where we retained certain methods or interpretations bring the paper up to a satisfactory level of quality. Please note that in some cases the comments have been re-arranged and grouped based on themes. This was done to facilitate an easier evaluation of our responses.

Reviewer #1:

Issues with data reporting and analysis

We thank Reviewer #1 for their detailed review and thoughtful comments. We have very carefully considered each point and note that these comments have led to a deeper exploration of the literature, that has provided us an opportunity to better explain our findings and expand on our interpretations. While in numerous cases we have not made the changes that were requested, we hope that it is clear that we have very carefully considered each point and chosen the approach that best suits the aims of our study.

The data set is extensive; however, I had a difficult time following the data, how they were obtained, the environmental setting and the implementation, resulting in multiple readings of the manuscript. I have a few issues (and questions) with the manuscript.

As detailed below, based on the feedback from both reviewers, we have made significant changes to the reporting of data collection, the use of data, and the results. We hope that these changes help Reviewer 1 follow the approach and findings that we've used in this study.

The data collection on the various lakes (inland waters) is vague and needs to be expanded. Specifically, the data for the individual lakes/ponds, their properties (depth, area, etc), ancillary data, dates of data collection. I believe these data must anyway be supplied should the paper be published.

We will supply these data for evaluation and now added a data availability section noting we will provide open access to the database that contains relevant information. We also note the

publications where existing data came from. We have revised the gas flux calculations section of the methods in great detail and note that now our use of these data comes from only three published sources (two articles and one thesis). The bulk of the paper relies on these data and they have also been extensively detailed in those publications.

One concern is that the lakes could be overturning while sampling occurred, which significantly affects surface methane, salinity and DOC – especially if the lake is experiencing an anoxic hypolimnion.

As detailed in the methods section, sampling was restricted (in a system that may have been stratified) to peak summer periods in the survey of primary sites, to eliminate issues such as the one raised here (effects of overturn). Therefore, we are as confident as we can be that this is not a confounding issue in the dataset.

In fact, the authors state that each lake was sampled 3 – 5 times, however no statistics or at least mean and STD are given.

For the subset of sites (peripheral wetland survey) where this sampling was conducted, all the statistical reporting can be found in supporting tables 3 and 4 and are discussed in supporting text S2. There were already extensively described in the previous version.

Of the 193 primary survey sites, they represent 1 sample per site in summer, so no statistics of this kind are presented.

For the ebullition data we have clarified now that we present repeated sampling from one location, which we now summarize in figure S9 where we have added error bars to the salinity versus ebullition rate figure seen in the paper (Fig. 2), but for which the error bars would be too much clutter to add to an already busy figure.

A final point, I think that rivers should be excluded in this analysis, as these are completely different systems than lakes and ponds (high flow rate, turnover, rapid surface gas exchange), etc.

Based on this comment and those from Reviewer 2, we remove rivers from the bulk of our analyses and title in this revised version of the paper. We did retain the rivers in only 2 key locations (table 1, and figure 1 a) because we feel it is important to present the differences between system types. In our opinion it is a strength of the paper that we explored these patterns across diverse ecosystem types, found distinct relationships, and present the correlation in figure 1b (accounting for DOG and salinity differences) that helps to explain the broad differences in the system types that may lead to the observed trends in figure 1a. The Reviewer is correct to point out the river systems are different, but we don't think this is a reason not to include the data. We feel that underreporting null results would be doing a disservice to the scientific community.

Line 81 – I am not clear why the authors present river systems in this paper, as these systems are substantially different than the ponds and lakes. I would suggest omitting the rivers from this study, and diving deeper into the fact that salinity affects CH₄ in small systems (i.e. ponds), and not lakes. It would be interesting to know where the bathymetric cutoff would be. They touch on this point in line 88, but think that this could be further explored.

We hope that in responding to this and the comments from Reviewer 2, that we have found the right balance between refocusing the paper on lentic systems, especially the small ones, but also presenting valuable information about the rivers that highlights these systems are indeed not a place where this salinity mechanism is relevant. We have detailed above all the steps that we have taken to deemphasize the rivers and are hoping that this addresses Reviewer 1's concerns. Unfortunately, we do not have extensive bathymetric information of most of these sites, and have considered such a question as an important point to explore in future research.

Figure 1b – Again, I strongly suggest removing rivers from this analysis. Overall, this correlation is very weak and I find this unconvincing.

We have removed rivers from the analyses as outlined above and below, but we have retained them in the paper simply to demonstrate that the relationship is indeed unconvincing as Reviewer 1 notes. We feel that it is important to communicate this basic message to readers, since there is great interest in the broader aquatic community to understand methane dynamics in rivers, and salinization is a topic that is highly relevant in rivers.

Issues with the use of salinity as a predictive metric

The fact that sulfate suppresses methane production (and thus fluxes) is well known (especially in the oceanographic community), but perhaps less known is the number of brackish (i.e. sulfate-rich) lakes – which the authors use in their estimates.

We respectfully point out that knowledge from oceanographic research and marine settings, though supplying limnologists with basic intuitive knowledge, is not in itself sufficient to understand and account for inland water biogeochemical processes. Oceanographers have indeed known about the relationship between methane and salinity for a long time (Reviewer 1 speaks specifically about sulfate, which we address in a separate point). In limnology, this is a very common intuition that many people have on many topics in our field. However, knowing about a possible relationship, and quantitatively demonstrating its importance in an entirely different ecosystem type (inland waters) at a different spatio-temporal scale (ecosystem to regional emissions) are two very different things. The latter has not been done for salinity and methane flux for inland waters at a broader regional or global scale. Herein lies the value of our paper, and the cutting-edge nature of our discoveries. We stated from the beginning and were up front about the fact that the fine-scale microbial processes have received attention, but nowhere has anybody fully accounted for the ecosystem-scale impacts of these mechanisms on emissions from inland waters in the ways that we have here. Had this been established, we would already see salinity or some equivalent integrated into the emissions models that people use (e.g., see Beaulieu et al. 2019, another recent and relevant paper on the topic in Nat. Comm). Generating these findings is a critical step that takes this topic from intuition and opinion into a quantitative realm.

As an extension of this last point, we note that salinity as we use it here is integrating more than sulfate alone (see below response to next point). This observation further distinguishes processes found here from marine work focusing on sulfate cycling alone.

Lastly, we also appreciate that Reviewer 1 has noted the novelty in our presentation of the new pond and wetland surface area dataset we use to scale emissions to the Canadian Prairies.

Surface water extent is indeed not well known and quantified here, so the use of this product is a major advance for national emissions estimates. Being the first paper to use this for emissions scaling, our paper is making a very important contribution to understanding emissions from the vast (>780,000 km²) Prairie Pothole Region.

However, the authors' key message that emission models should include salinity (that is sulfate, I disagree with using salinity as a parametrization here) could be a solid message, however, it is still unclear how this could be applied, and in which cases. Throughout the manuscript, I question the use of Salinity as this does not necessarily imply sulfate and could result in considerable error in the authors' correlations. In fact, they measured SO₄²⁻ in a lot of the study sites, so I do not understand why this was not used as the key parameter. My point, is salinity really an appropriate measure here?

We maintain that a focus on sulfate alone is a more narrow and incomplete exploration of the factors limiting methane production. In this sense, we appreciate this advice from Reviewer 1, but feel that it would be a sub-optimal exploration had we narrowed down to only looking at sulfate relationships to methane. Undoubtedly, these are questions that other readers would have, so it prompted us to compile an extensive supplement text section explaining the potential reasons why salinity is a broader, and more integrative predictor than any individual ion.

The assumption that sulfate alone is responsible for the patterns underlying salinity-methane trends worldwide is incorrect, and we now detail our reasoning in an extensive supplemental text section that we hope can alleviate the concerns raised in the first round of review. Importantly, salinity as a master variable is integrating more than DIRECT sulfate effects alone, and is indeed an important predictor of emissions patterns from hardwater systems at the broadest scale. As Reviewer 1 points out, sulfate is the best known and probably most important ion that sets redox conditions that favour alternate processes over methanogenesis. However, there are many other ions (nitrate, manganese, iron, organic matter etc) that are also interacting with the cycling of methane in complex ways (that we do not fully appreciate yet), and these ions change in availability with increasing salinity. We do not belabour the point here as it is detailed in the supplement. Instead, we present two figures that, as discussed in the supplement, provide clear and independent evidence that we hope the editorial board finds compelling. If, as stated as a reason behind the initial rejection, the salinity-methane connection is strictly dependent on sulfate, we would not see clear effects of other salt manipulations on methane. In reality, even NaCl additions alone induce complex biogeochemical effects that suppress net methane production in sediments (note that we use the term "net" because effects lowering the rate of methanogenesis, or enhancing anaerobic oxidation, or some mix of both effects could yield these observed trends). We present findings from two regions, distinct in their geochemical features from our study region, where non-SO₄²⁻-based manipulations yielded suppressed net methane flux:

[Redacted]

Example taken from NaCl additions and dilutions for lake sediments in Camacho et al. Water, 2017, 9, 659.

[Redacted]

Example taken from NaCl additions to wetland sediments in Baldwin et al. Wetlands, 2006, 26, 455-464.

All of this shows that a focus on sulfate alone is incomplete in the context of our study because it does not reflect the bigger picture effects of salinity on methane cycling that underly our empirical observations. We stress the point that this is not a question of which metric is the better one to predict methane cycling, but that using salinity (akin to other important metrics like total phosphorus concentrations or dissolved organic carbon content) captures the broadest relationships, while investigations using individual ions like sulfate provide finer-scale and complimentary information. In this context, the critique of our use of salinity is equivalent to criticizing a paper for using total phosphorus, total nitrogen, or other bulk chemical properties as predictors that establish global empirical relationships in place of metrics that capture a smaller subset of the element (e.g., requesting that we only use soluble reactive phosphorus, or ammonium or nitrate, or even aromatic carbon forms, to establish trends). We elaborate on this in the new supplement.

In summary, salinity is indeed reflecting distinct relationships between ions and methane that undoubtedly vary regionally or between system types. This is not a flaw and should not preclude using salinity to establish patterns at the broadest scales. We make it clear, we hope, that salinity is a useful metric in regions with diverse geochemistries. The two examples pasted above in response to comments from the Editor clearly demonstrate this point (we elaborate on this in the supplement). Further, our meta-analysis shows inverse relationships in all the regions where data exist, despite wide ranging geochemistries.

While the Fig S3 is somewhat convincing relating Log of sulfate to Log of salinity, since it is on a log-log scale I suspect there is substantial error here.

This is one strong indicator why sulfate alone is not as comprehensive as salinity as a predictor. This considerable variability in the SO₄/salinity relationship implies that using sulfate concentrations as predictor does not always reflect ecosystem salinity, even in our study region. The new figure S6 in the supplement showing that the use of SO₄ and salinity as predictors (standardized by amount of DOG) yields different correlation strengths is particularly important when we consider the agricultural ponds, where SO₄ alone doesn't reflect the underlying relationship between salinity and methane as well as in wetlands, but when combined salinity and SO₄ content yield comparable correlations at the larger scale. We have explained above and in the paper why this proposed use of SO₄ alone is a more narrow focus than what we wanted to take in the paper, though follow up work looking at individual ions is an important next step.

I understand it is the easiest to measure, but perhaps the actual electron acceptor that outcompetes the methanogens would be a much more solid analysis.

As we detail in the supplement, there is a lot more going on in hardwaters than a suppression of methanogenesis alone by direct effects of SO₄ concentration increases. Therefore, while this is obviously an important mechanism of suppression of rates of methanogenesis, it is not the only mechanism that leads to a restriction of net methane production in sediments. For these reasons, our goal of predicting methane trends in hardwaters at the broadest scale leads us to disagree with the reviewer statement that their proposed approach would be a more solid analysis.

My suggestion would be, including the above comments, include only data where sulfate was measured, and only include lakes and ponds. The fact that these two systems respond oppositely in terms of CH₄ to increasing SO₄²⁻ (though the correlation for CH₄ and SO₄²⁻ for lakes is very weak) deserves much more description and investigation.

Please see above responses to use of salinity versus sulfate more broadly. We agree it would be nice to expand on the contrast between small and large systems, though given the already-long manuscript and focus on other key issues that we feel take priority here, we believe that this important issue is one we need to follow up on in future research.

I question the use of CH₄ concentrations in these types of studies, as there are some many parameters that can affect concentration in the surface waters (wind exposure, oxidation rates, light penetration, etc). I think using rates (i.e. fluxes) is much more robust and revealing. This is a nice data set, and potentially nice story. Using only the waterbodies where CH₄ flux, concentrations

and SO₄²⁻ are all measured would make for an interesting, and very applicable study – especially when validating the data versus the methane modeling approaches.

The use of concentrations for some portions of the paper was deliberate because we did not want to introduce yet more noise in the patterns and broad comparisons through the need to account for yet more physical differences and potential errors. We feel that for the purpose of comparisons in figure 1, table 1, and associated supporting info., comparing concentrations is more appropriate.

However, we note that we are already extensively discussing emissions patterns for 139 of the 193 primary survey sites (in figures 2, 3, and in the scaling calculations later in the results/discussion), so we are essentially already doing what Reviewer #1 requests in the bulk of the paper. We note that exploration of emissions was restricted to these sections because we were more confident in the emissions calculations and data for this subset of the paper, and feel the results are stronger using this approach. Finally, Reviewer #1 mentions using SO₄²⁻ concentrations here, though we have explained our rationale for not doing so in detail above.

Line 42 – it is very well known that SO₄²⁻ inhibits CH₄ production. Check the oceanographic and coastal research literature.

We do not dispute this statement from Reviewer 1. In fact, we cited important marine and coastal papers in this sentence, as well as inland water papers, one of which clearly demonstrates that another form of salinity (NaCl) also suppresses methane production. To make sure that other readers do not take the sentence as a claim that the inhibition of methane production is not well established, we've reworded that sentence to say that there "is clear evidence". However, as noted above, there is a very important difference between intuitively assuming that a mechanism demonstrated in marine habitats extends to inland waters and demonstrating ecosystem to regional scale ramifications of such a process. The sentence immediately following the one highlighted here emphasizes this knowledge gap and makes it clear where oceanographic knowledge reaches its limits of application.

Other specific comments

The use of "core" study site is ambiguous, as I often think of sediment core. Suggest changing to "primary".

We agree and have changed this terminology throughout the paper.

Line 40 – it would be interesting (if possible) to know how salt-rich lakes are currently represented proportionately in the database used to estimate current global estimates of inland water emissions.

We agree and were not able to find such data associated with past research. This is something that the research community will hopefully establish once our work is publicly available to stimulate interest in this topic.

Line 50 – you list SO₄²⁻ and Fe³⁺ as energetically favorable electron acceptors. How important are other EAs in these systems? The relevant ones should be listed here.

We now elaborate on this quite extensively in the supplemental text S1. We hope that this provides the context needed to understand and appreciate the issue.

Line 60 – the authors survey 193 freshwater systems. As a reader, I have an issue interpreting these results without more information on the individual waterbodies.

We hope that the provision of the data file will give readers the context needed to understand the features of individual water bodies (see data availability statement after main text). We have revised the methods section to make it clear how we subset the data for the use of sites with emission data and make more clear reference to the fact that three detailed publications have already elaborated on these study systems. We hope that this clarifies the issue.

Line 78 – define “full suite of limnological parameters”.

We now present table 1 immediately after this paragraph, which provides a list of the parameters included in models.

Line 94 – It would be helpful to the reader if the authors could present upfront the global representation of wetlands and ponds in general, and the fraction that are brackish – or better yet, the ones with elevated SO₄²⁻. A table here quantifying the global setting would be appropriate.

We are unaware of such information that we could present here and note that the global extent of such small systems is an active area of research, with wide ranging estimates of area that are error prone. We did add to the introduction half a sentence noting hardwater systems make up a fifth of the global inland water surface area. We also hope that other changes we have made give some context on the geochemical features/differences among global regions. We now added a supporting figure detailing the relationship between SO₄²⁻ and Cl⁻ in diverse hardwater systems from around the globe (Fig. S10) and discuss in detail in the supporting text S1 the differences in SO₄²⁻ to Cl⁻ concentration ratios in our region versus other regions. We also cite literature sources detailing metal and trace metal content along salinity gradients. We hope that these revisions provide more global context on aquatic geochemical differences.

Line 100 – why not show the relationship between CH₄ and TN/TP ratios? This can easily be added to the supplemental.

We have added the relationship between methane and both TN and TP to the supplement as fig. S3. We think that the comparison to each nutrient concentration, rather than the ratio of the two nutrients, is a better indicator of trophic state, so have taken a slightly modified form of this advice to hopefully clarify this section.

Line 204 – the authors state “Existing empirical models” are used to compare the model results with their measured data. As this is such a key component of the study, I suggest that the model details be presented so that the reader doesn’t have to search for them in the literature.

We have added a more comprehensive explanation of the models and our use of them in the methods section (see ‘statistical analyses’ section).

Line 208 the model is driven by chlorophyll a, I would be curious how the data presented here correlate with CHLa – could this be a main driver?

We agree with Reviewer #1 that the model using chla as a predictor assumes that system productivity or trophic status is the main driver of ebullitive emissions. This seems to work well in other systems, though clearly doesn't hold here. We agree that the lack of relationship with chla and CH4 is an important reason for the differences in observed and predicted emissions in figure 3. We felt that this was pretty clear in the manuscript but present the figure summarizing chla versus average ebullitive flux here for the interest of Reviewer #1. As the figure indicates, the patterns are not clear, though there may almost be a negative relationship if we ignore a few points. At any rate we do not want to overinterpret the figure. We would consider adding it if Reviewer #1 feels it is important, though held off as we already have quite an extensive supporting information file.

[Redacted]

Confirm that tables and figures (including supplement) are listed in order.

We have gone through the paper and made edits to the figures, tales, and supplemental materials. We hope there are no mistakes now and will further correct any potential oversights if they exist.

Figure 1a – This figure is very hard to read. As this is a key figure for the message of the paper, I make the following suggestions. Remove the River data (from the study in general), and present the lake, ponds and wetland data in individual panels with the data behind (mean and std for each site). The correlation for the wetlands and ponds is excellent – it would be great to see with the actual data.

While we have elected to keep this figure as is, we hope we have addressed this issue in several other ways. We keep this figure as is because we feel it is important to show the differences between the systems as a first step (we explain why it is important to show the 'null results' above). Yet now we first introduce the global multiple regression equations for each system type and the

combined lentic systems in the new Table 1. This is the global context and equations needed to understand the marginal effects derived from multiple regressions in figure 1a. We make specific reference in the caption to table 1, so readers should be able to draw a direct line between the two. We also point readers to figure S2 for a visual summary of regression equations to help those who like visual learning over tables. We have also added figure S6, which breaks apart these relationships by individual system type and when combined, because we agree with Reviewer 1 that it is nice to show the data behind the regressions. We hope that this addresses these concerns.

It is interesting (while very weakly correlated) why the methane increases with salinity in the lakes. In my opinion, this deserves a more detailed focus within this paper, as it is contrary to the authors' key hypothesis. That is, if the salinity decreases the methane in ponds and wetlands, is this compensated by an increase in methane production with salinity in lakes?

As we mention above, we have devoted the bulk of the paper to detailing relationships in the systems where these trends are most important. While the differences between large and small lentic systems is interesting, we feel that to properly explore this issue would require a different study design that is not a great match to this dataset and study design (specifically planned to explore the themes we focus on here). To explore the topic that Reviewer 1 proposes, we would need to follow up with a more sophisticated and detailed survey of methane dynamics across depth transects of systems of increasing depth or size.

Esp. including all the different water types in a single correlation. I suggest developing the individual correlation for each waterbody type. This correlation however is the basis for the model shown in Fig. 1c. I question if this can be applied in the manner it has been here. For the first thing, since Fig. 1a shows an inverse relationship between salinity and methane, I am unclear how these data can be used to infer the model. The separate contributions of DOC and salinity in the model in Fig. 1c could be better explained.

As per comments from both reviewers. Figure 1c now does not include all system types and we exclude rivers in all subsequent analyses. We hope that the removal of rivers throughout the rest of the paper, the inclusion of regression statistics in the new table 1, and addition of figure S6 provides the context Reviewer 1 is looking for.

Figure S2. This figure shows the effect size and the confidence of the regression analysis on the various parameters driving the concentrations of dissolved pCH₄. It seems that the salinity only negatively impacts methane in the small lentic systems. While the effect size is shown to be large for lake (salinity) methane, it seems to be neutral which I don't fully understand.

Please note that we have revised figure S2 following other comments and have removed the rivers from this analysis, and focus on the individual lentic systems, and small lentic systems combined that are now the focus of figure 1c. We have also added figure S6 to show the data behind the individual and combined lentic system relationships that we focus on in figure 1. Reviewer #1 noted that the salinity effect is negative in the small systems. We now emphasize this in the paper. For the larger lakes, this relationship is non-significant, so while the salinity relationship with methane is negative, it is non-significant. This non-significance is reflected in the use of the dashed line and massive error bars in figure 1a for larger lakes, and we have now placed table 1 at the

beginning of the results/discussion section to better guide readers through the interpretation of figure 1. In table 1 we show that the larger lakes model is non-significant, so this should hopefully provide the context needed to understand figure S2.

The strongest effect is the area – which I feel should be discussed in more detail. Statistics aren't my strong point, but I have trouble understanding what this figure actually means. Also, in this figure, the salinity only really seems to have a strong effect on small lentic systems. Also, DOC seems to have the largest (positive) effect on methane, but this is not seen in the individual class systems (i.e. lakes vs rivers vs small lentic). How is this explained?

Here, as we have mentioned in response to other comments above, the multiple regression for larger lakes is nonsignificant overall, so we are hesitant to say too much about the relationship between pCH₄ and surface area. We agree that the salinity effect is restricted to small systems and have revised the paper to emphasize this point. We have also explored the patterns with DOC in more focused detail by system type and focus, as requested, on the small lentic systems (see supporting text figures S3 and S6, and revised figure 1b). We discuss the system differences in detail at the beginning of the results & discussion. We hope that these changes help to clarify the issues raised by Reviewer #1.

Methods – Sampling time: How does that effect outcome due to changing climate (June – August). Sites were sampled 3-5 times, it would be useful to know the range of values and variability in salinity (i.e. sulfate), pCH₄ and DOC during these times.

We agree that sampling time in general is an important consideration, and this is precisely why we conducted such an extensive analysis of the variance associated with our peripheral wetland survey sites (summarized in supporting information text S2). This supplement has already summarized our evaluation of the importance of multiple sampling per site, timescales (monthly to less frequent, i.e., seasonally,), and inter-regional spatial scales, in terms of inducing variability in our dataset for pCH₄. This analysis demonstrates very clearly that by capturing the long environmental gradients reflected in our inter-regional survey, we capture most of the variance in pCH₄ that may be induced by wide ranging environmental conditions, and that within-site variations in conditions, or those induced by shorter-term variation, are much less important than the spatial gradients that we capture with the survey. We are therefore as confident as reasonably possible that a compilation of summer samples is a reasonable way to evaluate the drivers of methane cycling in hardwaters, even with the comparatively small ranges in environmental conditions that may be induced with a June-August sampling window. We now note that it was the peripheral dataset that was sampled 3-5 times (for the purpose of this analysis), so that readers are not confused with the primary survey sites.

Methods – Line 147, which type of multiparameter probe? Only surface, or were profiles used?

We now note that we use surface measurements in our analyses, and that “The probes used at each site varied in model, but were all either Yellow Springs Instruments (YSI) or Hydrolab DS5 model probes, and all were routinely calibrated before sampling.”

Methods – Line 393, how was the wind measured at each site? What is the justification for using K600 parameterization from Cole and Caraco, 1998? That study used tracers to estimate k600, and has since been shown to underestimate k600. My suggestion, since the authors performed direct chamber measurements on 10 ponds, calculate the k600 directly from their data and compare with other parameterizations that may be better suited, e.g. MacIntyre et al. 2010 Geo Res Lett.

Please note that we have revised this section to more accurately reflect the calculation of flux rates. In essence, we use only published and already detailed emissions rates that were derived from chamber flux measurements, averaged across sites and time, and applied to all ponds and wetlands (as reported in Webb et al. 2019 Biogeosci, and Jensen et al. 2023 JGR-B). As these data were already reviewed and published, we are hesitant to recalculate emissions rates with other models that each have their own strengths and limitations. For instance, earlier work has shown that for a small to mid sized lake, averaging chamber based CH₄ flux estimates made over 24hr periods yields near identical K600 estimates as using averaged wind speed data in the Cole and Caraco 1998 model (Bogard et al. 2014 Nat Comm), so there are clearly instances where the Cole and Caraco model works extremely well. Plenty of other examples like this exist in the literature, so given the wide range in potential model options and related outcomes, we are hesitant to go in the direction of comparing gas transfer models, as this really is not the focus of the paper and would lead to an overly complicated and off-topic discussion.

Methods – Line 398, The authors should provide more details on how bubble ebullition was measured. How were the instruments (funnels) deployed to ensure that they did not disturb the underlying sediments? It would be interesting to see the maps of the ponds, depths of deployment and data.

We now clarify bubble trap details in methods section on estimating flux rates. We also mention where extensive information on the sites has already been published (Jensen et al. 2023 JGR-B), and where the ebullition data were initially reported (Jensen M.Sc. thesis). We note that Jensen et al. (2023) provide considerable information on the sites, including the map and example photos that we provide below.

[Redacted]

Figure from Jensen et al. 2023 JGR-B. This paper reports the diffusive emissions data from the 5 wetlands and 5 agricultural reservoirs (ponds) included in our paper, plus an additional 30.

Reviewer #2:

The biggest issue I have with this study is how all different aquatic systems (rivers, lakes, ponds and wetlands) are combined in the analysis. The breadth of systems is a strength of the manuscript, but the way it is used may not be fully accurate. This is because in some systems (rivers and large lakes) salinity is not that important relative to other factors according to their analysis, and therefore the conclusions and subsequent analysis should reflect that. The title “Salinity causes widespread restriction of methane emissions from inland waters” promises that salinity is THE driver across all systems and is widespread, while the analysis doesn’t fully support it. I will pull a couple of examples to better illustrate the problem, and then suggest a way to address it.

We thank both Reviewers for raising this point and have revised the paper to focus on small systems. We explain above why we retain river and large lake data in select locations but note that the vast majority of the paper has now been focused to patterns in small lentic systems.

In the opening sentence of the results and discussion (L79) states: “Across all core sites where the full suite of limnological parameters were measured, salinity had a strong negative effect on pCH₄ in a multiple linear regression that also included temperature^{33–35}, and common proxies for organic substrate (DOC and TP_{4,5,36,37}) (Table S2 and Fig. S2)”. Here I am not sure if they refer to the multiple linear regression including all data, or the system specific linear regression. If the former, I am not sure is statistically sound to include all these different aquatic systems in the same analysis, without including the system as a factor in the model. If the latter, it is not correct given that the MLR for lakes and rivers show in fact a positive effect of salinity on methane emissions (table S2).

We have now revised this analysis to remove the combined regression, so this comment is generally not applicable to the current version of the paper anymore. To make it more clear to readers what we are presenting, we have now included table 1 (initially in the supplement) and explain it in the text before moving on to figure 1. We have revised table 1, figure 1b,c (relying on the regression model), and supplemental materials all so that the rivers and ‘global MLR model’ presentations are removed from the bulk of the paper. We retain rivers in figure 1a for reasons outlined in response to Reviewer #1 comments.

In the important analysis of Figure 1. Is the regression in panel b on all data, including big lakes and rivers? If so, is it meaningful when they behave so different than ponds in wetlands? The analysis in panel c depends on the relationship in panel b, and given the conflicting patterns between lakes/rivers vs ponds/wetlands (shown clearly as the marginal effects in panel a), it may be a bit misleading to use all data to build a model of DOC/salinity vs CH₄, which is then used to do some nice and clean lines of salinity and CH₄ across DOC (panel c). I don’t dispute these results, but that it may not hold as cleanly for rivers/lakes as for the other systems based on the data provided.

We agree with the assessment here. We keep rivers in panel 1a (and table 1) because we feel it is important to first point out that these differences appear to exist between system types. This is a very novel observation even if it is not the core focus of the rest of the paper. We have revised figure 1b,c based on this comment so that it only represents patterns for the small lentic systems.

The authors are clear about the fact that salinity is not a first-order control across all systems (lines 83-90), show the data in the multiple linear regressions (Table and figure S2) and the marginal effects in figure 1a. But then in some other cases the authors use all data lumped together to do some broader analysis and claims, that to the reader it may appear that they are applicative across all inland waters while they are not for larger lakes and rivers (again based on this dataset). I would suggest two ways to address it: Solution 1: The authors have done a much more extensive effort on the aquatic systems that have a strong effect of salinity, with a regional survey of ponds and some eddy covariance data for wetlands, great work! Given this, maybe the work should have an initial assessment across all systems as follows: “we sampled all those systems, seems like the effect of salinity is stronger in wetlands and ponds because X, in lakes and rivers its non-significant and even positive because of Y. We then proceed with a detailed analysis and landscape relevance of salinity controlling CH₄ emissions in ponds and wetlands”. This would still provide a

compelling work and with a bit more rigour, maybe you should leverage in the introduction then that across all inland waters, wetlands are the largest source of CH₄ and that small ponds the most uncertain one. Solution 2: Use literature data to expand the analysis for lakes and rivers. There is a large new database of river methane data that includes conductivity data (<https://essd.copernicus.org/preprints/essd-2022-346/>), and a new global estimate of lakes (<https://agupubs.onlinelibrary.wiley.com/doi/full/10.1029/2022JG006793>) and reservoirs (<https://doi.org/10.1029/2021JG006305>) with supplementary data. This would be more work and I am not sure the lentic products have salinity data, but it could be crossed with other chemical databases. This would be more work, but if you prefer to maintain the focus across all inland waters more data is needed to support the role of salinity in other aquatic systems besides ponds and wetlands.

We thank Reviewer #2 for taking the time to note a few potential solutions to the issue. We have opted generally for solution #1, as we have laid out in our responses above to comments from both Reviewers. We note that we have already summarized in the text that we saw between-system type differences, but improve this by making the transition from the all-system comparison (now figure 1 a) to the focal systems (small lentic, now figures 1b,c, 2 and on). We now state at the start of that section that we focus on the small lentic systems where salinity appears to have the most important effect on these systems (based on fig 1a). We modified the figure caption to reflect this.

Specific comments

L 41: Instead of freshwater volume, what is relevant in this article is the surface area from an emission perspective.

Added reference to Duarte. Et al. 2008 JGR-B and note that they make up ~20% of inland water surface area. Obviously, the uncertainties linked to the area of small systems (as this Reviewer has noted) should be considered here, so this is what we say it is approximate.

L60: Here the number of sampled sites is a bit confusing, I would rephrase to: “Here we survey 193 aquatic systems, including rivers, lakes, open-water wetlands, and agricultural ponds and spanning...”.

Changed as suggested.

L94: Maybe worth noting that small lentic water bodies are the largest source of uncertainty in aquatic methane emissions, both in rates and the total global surface area.

Changed as suggested.

-L113: Here is unclear what aquatic systems you are speaking about. Is it all of them? Ponds?

We now make it clear that it is the small lentic systems by adding a header here and revising text. Subsection is now “Interplay between salinity and OM regulates pCH₄ in small lentic systems”.

L124-L133: This paragraph is quite short and technical. Maybe it can be clarified a bit. Two suggestions here: L128: Instead of giving specific changes in ppm of CH₄, all of that could be

simpler by saying “For example, at the same DOC concentration (10 mgL), a 0.5 ppt increase in salinity at low salinity conditions (0.5 ppt) results in a 50% reduction of CH₄, while a similar increase in modest salinities (4 ppt) only decreased CH₄ by 11% (Figure 1c)”.

Then at the end is missing a take home message. For example a quick draft of one: “Thus, increased salinity in solute poor systems may have a stronger effect on methane dynamics than in already saline systems, given anthropogenic pressures that result in salinization of freshwater systems (ref) but also in an increased supply of OM and nutrients, the net balance may not be an increase of methane emissions as previously anticipated (ref).”

We have modified this section as suggested. Please note that the simulation is now based on model output from the revised fig. 1b, so values have changed slightly, but the same general observations and interpretation remain.

-L150: Ecosystems means ponds here right?

Ponds and wetlands. 5 each. Clarified now.

L176: You could even say that the low salinity wetland has three times more DOC than the high one to further support this result. Take it or leave it.

This is a nice observation, though we decided to leave it as is and will dig into the details in a follow up paper.

L199: For the Mexican pond data, If 0, how is it shown in the plot and the regression line calculated? Also, there is a study of Spanish saline lakes that could be worth to include in this figure <https://www.mdpi.com/2073-4441/9/9/659>

For the Mexican ponds, we make this line a dashed one to make it clear it is not significant. For the study cited here, we had already included the paper as a reference in the text and note that the data reported in this paper are only for sediment incubations (not ambient data from surface waters), so unfortunately, we cannot add this since it is inconsistent with our other systems. To our knowledge we are unaware of companion papers that may include such measurements from these lakes.

L204: Here and in figure 3, it is all based on published empirical models but these models are not presented in the results. For transparency, I would include the models in the methods, and how they perform in the supplement materials. These results further strengthen my main issue above, given that the model for rivers does OK (with an offset, but these are different systems from the model) and also for lakes. This analysis would be stronger and more compelling if restricted to ponds and wetlands.

We now report the models used in the methods section (statistical analysis subsection). We have restricted this to the lentic systems as well. We were not quite sure what Reviewer #2 was requesting regarding model performance, since the statistics and residual deviation etc. is provided in the paper. We would gladly provide more information if needed, with more specific requests.

L177: The regional upscaling is sound and compelling. In the global attempt though, the authors are cautious enough so it is ok, however I find more interesting to highlight that both the area and total emissions are heavily uncertain for ponds, and this study provides an important mechanism that could refine global estimates. This is because global estimates from most inland waters are still quite bad from a mechanistic perspective, so maybe no need yet to quantify biases if salinity is not included.

We agree with the comment raised here by Reviewer #2 and think that we have captured this general interpretation in later sections of the paper where we discuss implications of our observations. For instance, where we state “Therefore, as a first-order approximation, our calculation demonstrates the importance of obtaining accurate emissions data for Prairie ecosystems in the context of our national GHG emissions inventory. Inaccuracies in these calculations have far reaching implications for national emissions mitigation, and how aquatic ecosystems are represented in these budgets. The inclusion of salinity represents a simple but major refinement to estimates of aquatic emissions from hardwater ecosystems in Canada and likely other nations (Fig. 2)”. We think that this captures the spirit of the comment but can revise as needed with further guidance.

L342: Does eddy covariance need to be in caps?

Changed as suggested.

L 358: Would be helpful to know what was the norm and the exception. Any number to say what % of sites that were filtered or not?

We now report that we calculated total nutrients for 105 sites.

L367: links or reference to the data sources?

Data availability statement now added at end of main text. The final datasets will be uploaded once the paper is accepted, and no further revisions are required (to the FRDR, which is routinely used for all Bogard lab publications).

L369: While still being in Canada, that relationship is from another biome that changes those hydraulic scaling relationships. I would rather use a larger assessment such as Raymond et al 2012 L&O Fluids and Environments (<https://doi.org/10.1215/21573689-1597669>)

Rivers have been removed from this portion of the paper, so comment no longer applicable.

L392: I am not a wetland person, but is it common to use a lake based model of gas exchange for wetlands?

Please note that we have revised this gas flux calculation section to reflect the use of chamber-derived k600 values. In response to Reviewer 1 comment above, we also provide some context about choosing different models of k600 in the context of this study.

Table S2: Would be interesting to see here whether each given parameter is significant or not in the regression. Instead of using gray formatting for the whole equation, it could be used for that, or using an asterisk .

Please note that this is now table 1. Given the small number of sites in this portion of the dataset, we hesitate to read too far into the relationships underlying each predictor (as we explained in several responses to Reviewer #1). Both sets of comments are indeed issues that interest us as follow up questions/explorations.

Now seeing fig S1 I see that most rivers sampled are in Alberta, could there be some regional patterns here of salinity in inland waters, rather than a system difference? Just wondering, no need to do a full analysis on this.

We agree that this is an interesting and outstanding question. We will follow up on this in future work because we do not know at present whether some regional effect is lurking that alters the relationships in rivers. This is doubtful, and they probably are quite different from lentic systems (the relationships probably would hold with more data). But we need to explore this in future work.

Figure S2: Typo in “Small lent.” Top right corner

Changed as suggested.

REVIEWERS' COMMENTS

Reviewer #1 (Remarks to the Author):

Evaluation of the Revised Manuscript Titled “Salinity Causes Widespread Restriction of Methane Emissions from Small Inland Waters” by C. Soued, M. Bogard, et al. submitted for publication consideration to Nature Communications.

I've carefully reviewed the updated manuscript and the authors' responses to the comments from the two reviewers. Overall, I believe the authors have effectively addressed the reviewers' concerns. The narrative is now much clearer. I offer two additional suggestions for consideration, but the final decision on their inclusion remains with the editor and the authors.

Data Availability: I'm uncertain about Nature Communications' data availability policy. However, I understand that it typically mandates that readers should be able to replicate the procedures detailed in the manuscript. To facilitate this, I suggest that all data utilized in the authors' analyses be made accessible in a repository. This would enable the reproduction of the manuscript's calculations and plots. For our publications, we provided all data used in plots, calculations, etc., in an accessible database with a DOI. This promotes transparency and prevents others from having to search through publications for these data.

Inclusion of River Systems: I noted that the authors have chosen to retain some river data, specifically in Table 1 and Figure 2. However, I reiterate that I believe the clarity and focus of the manuscript would be enhanced if the river data were omitted entirely. Its presence detracts somewhat from the core message. Nonetheless, the ultimate decision rests with the editor(s) and the authors.

Reviewer #2 (Remarks to the Author):

The authors have successfully addressed all issues raised by me and the other reviewer, and the manuscript has substantially improved on what was already a great work. I have read through all materials and found just a couple of small things. The new text S2 is a bit lengthy and maybe an overkill, but is a useful content to have regardless.

Below a couple of typos I found:

-L171: Maybe is clearer saying “in 5 ponds and 5 wetlands”

-L1082-1084. Here with the superscripts for chemical elements, the citations are confusing. Rather present the citations as (ref. xx)

Responses to Reviewer Comments

Please note that below we provide responses to reviewer comments in blue, italicized text, immediately after the initial comment, listed in standard black text.

Reviewer Comments:

We thank both reviewers again for another careful reading of our paper.

Reviewer #1:

Data Availability: I'm uncertain about Nature Communications' data availability policy. However, I understand that it typically mandates that readers should be able to replicate the procedures detailed in the manuscript. To facilitate this, I suggest that all data utilized in the authors' analyses be made accessible in a repository. This would enable the reproduction of the manuscript's calculations and plots. For our publications, we provided all data used in plots, calculations, etc., in an accessible database with a DOI. This promotes transparency and prevents others from having to search through publications for these data.

Please see the revised data availability statement – all data are accessible.

Inclusion of River Systems: I noted that the authors have chosen to retain some river data, specifically in Table 1 and Figure 2. However, I reiterate that I believe the clarity and focus of the manuscript would be enhanced if the river data were omitted entirely. Its presence detracts somewhat from the core message. Nonetheless, the ultimate decision rests with the editor(s) and the authors.

This is an entirely fair point that the Reviewer makes. We have opted to keep this line of inquiry because we feel that it is the best way to foster open science and avoid contributing to a positivity bias in the field. While the story may be cleaner without the river data included, we believe that readers should see the river results, even though they are non-significant with respect to the methane-salinity relationship. This is also the first exploration of its kind for rivers (to our knowledge), so we feel that this is important knowledge that others will likely find useful. We deem the pros to outweigh the cons here, so have elected to keep the data in the paper.

Reviewer #2:

Below a couple of typos I found:

-L171: Maybe is clearer saying "in 5 ponds and 5 wetlands"

Changed as suggested.

-L1082-1084. Here with the superscripts for chemical elements, the citations are confusing. Rather present the citations as (ref. xx)

Changed as suggested.